# O-GlcNAcylated p53 in the liver modulates hepatic glucose production

Maria J. Gonzalez-Rellan [1,2], Marcos F. Fondevila [1,2,21], Uxia Fernandez [1,2,21], Amaia Rodríguez [2,3], Marta Varela-Rey[4,5], Christelle Veyrat-Durebex [6,7], Samuel Seoane[1], Ganeko Bernardo [8,9], Fernando Lopitz-Otsoa [8], David Fernández-Ramos[5,8], Jon Bilbao[8], Cristina Iglesias[1], Eva Novoa[1], Cristina Ameneiro[1], Ana Senra[1], Daniel Beiroa[1], Juan Cuñarro[1], Maria DP Chantada-Vazquez[10], Maria Garcia-Vence[10], Susana B. Bravo[10], Natalia Da Silva Lima[1], Begoña Porteiro[1], Carmen Carneiro[1], Anxo Vidal[1], Sulay Tovar [1], Timo D. Müller[11,12], Johan Ferno[13], Diana Guallar [1], Miguel Fidalgo [1], Guadalupe Sabio[14], Stephan Herzig [15], Won Ho Yang[16], Jin Won Cho[16], Maria Luz Martinez-Chantar [4], Roman Perez-Fernandez [1], Miguel López [1], Carlos Dieguez[1], Jose M. Mato [6,7,9], Oscar Millet[6,8], Roberto Coppari[5], Ashwin Woodhoo [17,18,19,20], Gema Fruhbeck [2,3] & Ruben Nogueiras [1,2,20 ✉]

p53 regulates several signaling pathways to maintain the metabolic homeostasis of cells and modulates the cellular response to stress. Deficiency or excess of nutrients causes cellular metabolic stress, and we hypothesized that p53 could be linked to glucose maintenance. We show here that upon starvation hepatic p53 is stabilized by O-GlcNAcylation and plays an essential role in the physiological regulation of glucose homeostasis. More specifically, p53 binds to PCK1 promoter and regulates its transcriptional activation, thereby controlling hepatic glucose production. Mice lacking p53 in the liver show a reduced gluconeogenic response during calorie restriction. Glucagon, adrenaline and glucocorticoids augment protein levels of p53, and administration of these hormones to p53 deficient human hepatocytes and to liver-specific p53 deficient mice fails to increase glucose levels. Moreover, insulin decreases p53 levels, and over-expression of p53 impairs insulin sensitivity. Finally, protein levels of p53, as well as genes responsible of O-GlcNAcylation are elevated in the liver of type 2 diabetic patients and positively correlate with glucose and HOMA-IR. Overall these results indicate that the O-GlcNAcylation of p53 plays an unsuspected key role regulating in vivo glucose homeostasis.

A full list of author affiliations appears at the end of the paper.

More than half of all human tumors contain mutations or deletions of p53, and thereby, p53 is an intensively studied protein primarily as a tumor suppressor[1]. Besides cell proliferation, p53 also plays an important role in a large variety of biological functions[2,3]. The role of p53 in glucose metabolism has mainly been assessed in cancer cells. In that setting, p53 reduces glycolytic flux by different mechanisms, such as limiting glucose uptake[4], inducing glycolysis and apoptosis regulation[5], upregulating the Parkin gene[6], or inhibiting the pentose phosphate pathway[7]. Despite all the knowledge in cancer cells, the function of p53 in glucose metabolism under normal conditions has been poorly investigated.

There is nowadays growing evidence demonstrating that cell cycle regulators have important actions in metabolic control[8–10]. This dual role for these genes is somehow expected because during tumor progression, cancer cells require high amounts of lipids, glucose, amino acids, and other macromolecules to construct new cells, and therefore, adaptive mechanisms to cope with large amounts of energy are required. Data gleaned in recent years have shown a positive association between p53 and the expression of gluconeogenic genes or glucose levels[11–13]. p53 is highly expressed in adipocytes of obese leptin-deficient mice[14] and after excessive calorie intake[15], while the inhibition of p53 activity in adipose tissue improves insulin sensitivity in mice[15]. Under diabetogenic conditions, p53 is upregulated in pancreatic beta cells and inhibition of p53 alleviates defective glucose-stimulated insulin secretion in diabetic islets[16]. Nevertheless, the relevance of endogenous hepatic p53 in the physiological fluctuation of gluconeogenesis, as well as the molecular pathways mediating these effects, remains totally unknown.

In this work, we report that hepatic p53 is stabilized after fasting and glucose plays a fundamental role in p53 stabilization that is dependent of O-GlcNAcylation, a dynamic posttranslational modification consisting of the addition of a single N-acetylglucosamine sugar to serine and threonine residues[17]. At the functional level, p53 binds to the promoter of PCK1, which is essential to maintain gluconeogenesis. We find that mice lacking p53 in the liver do not maintain blood glucose levels during caloric restriction; and mice and cells without p53 do not respond to the hyperglycemic effect of glucagon, adrenaline, and glucocorticoids. Moreover, mice and human hepatocytes over-expressing p53 do not respond to insulin-induced suppression of gluconeogenesis. Finally, in patients with type 2 diabetes (T2D), hepatic p53 levels are elevated and positively correlated with insulin resistance. Our findings demonstrate that p53 is a key player in the physiological regulation of gluconeogenesis in non-tumoral cells and further indicate that p53 is of clinical relevance for adequate glucose metabolism.

## Results

### Nutritional availability regulates hepatic p53 stability via O-GlcNAc.
Mice subjected to 24 h fasting show increased p53 protein levels in the liver, which return to baseline after refeeding (Fig. 1a). This upregulation is observed already 6 h after fasting and is maintained for 12 and 24 h (Fig. 1b). Intriguingly, p53 mRNA expression does not change during fasting (Fig. 1b), suggesting that the regulation of hepatic p53 by nutrient availability is mediated at the posttranscriptional level. Since leptin is a key metabolic hormone strongly regulated by caloric excess (inducing hyperleptinemia) or deficiency (inducing hypoleptinemia) we evaluated its possible role as a modulator of p53. However, p53 protein levels in fasted mice treated with leptin remain augmented (Supplementary Fig. 1a, b). Given that circulating glucose is also severely affected by energy intake, we fasted mice for 24 h and provided one group

of mice with sugar. As previously reported[18], fasted mice receiving sugar lose the same weight as the non-sugar supplemented fasted mice, but their blood glucose levels are similar to mice fed ad libitum (Fig. 1c). The fasting-induced increase of hepatic p53 protein levels is completely blunted in mice receiving sugar, while p53 mRNA expression remains unchanged (Fig. 1c). To examine whether the osmotic pressure could be affecting p53 levels, we administered mannitol, which has a potent osmotic diuretic action, to mice fasted for 24 h. As previously shown, fasting increased p53 and fasted mice treated with mannitol still show higher p53 levels (Supplementary Fig. 1c). The results obtained in mice were corroborated in human hepatic THLE-2 cells. p53 protein levels are increased in starved cells, and glucose supplementation inhibits this increase in a dose-dependent manner (Fig. 1d). To note, increased p53 levels in starved cells were maintained when these cells were incubated with mannitol (Supplementary Fig. 1d). These data collectively suggest that hepatic protein levels of p53 are negatively regulated by glucose and not by changes in osmotic pressure related to glucose levels.

p53 function is tightly controlled by its protein stability. Thus, we next aimed to identify posttranscriptional modifications that may be potentially responsible for the fasting-induced rise in p53 protein levels. Data obtained in C. elegans have shown that O-GlcNAcylation plays critical roles in similar events to those reported for p53 such as transcription, cell cycle, and cellular stress response[19]. In addition, O-linked β-N-acetylglucosamine (O-GlcNAc) modification is known to regulate and stabilize p53[20], and this posttranslational modification is dependent on intracellular glucose availability[21]. O-GlcNAcylation is affected by nutrient changes because its precursor, uridine diphosphate N-acetylglucosamine (UDP-GlcNAc), is positioned at the nexus of the metabolic pathways of glucose, fatty acids, nucleic acids, and nitrogen[22]. In addition to being dependent on nutrient availability, O-GlcNAcylation signaling is highly sensitive to various forms of cellular stress, including heat shock, hypoxia, and nutrient deprivation. In this regard, it has been consistently observed an increase in cellular O-GlcNAcylation of specific proteins in conditions of nutrient deprivation[23–25]. The addition and removal of O-GlcNAc depends on the two enzymes O-GlcNAc transferase (OGT), catalyzing the addition of the O-GlcNAc moiety from the high-energy donor UDP-GlcNAc, and O-GlcNAcase (OGA), catalyzing the hydrolytic removal of the sugar moiety from proteins respectively[21]. Since O-GlcNAcylation during nutrient deprivation is highly influenced by OGT activity, we assessed OGT protein levels in starved cells (Fig. 1d) and in mice upon fasting, where we observed a significant increase in OGT protein levels as early as 6 h of fasting, remaining elevated after 12, 16, and 24 h of fasting (Fig. 1e). Taken together, these data prompted us to assess in depth whether O-GlcNAcylation was involved in the effects of p53 in glucose homeostasis. Co-immunoprecipitation analysis shows increased O-GlcNAc in p53 in the liver of fasted mice (Fig. 1f) and starved cells (Fig. 1g). Thus, we treated THLE-2 cells with O-(2-acetamido-2-deoxy-D-glucopyranosylidene)-amino-N-phenylcarbamate (PUGNAc), an inhibitor of OGA that increases protein O-GlcNAc levels[26]. PUGNAc increased p53 protein levels in a dose-dependent manner (Fig. 1h). Furthermore, THLE-2 cells were cultured in a glucose-free medium and treated with OSMI-1, an inhibitor of OGT that decreases protein O-GlcNAc levels[27]. The treatment with OSMI-1 reduced fasting-induced p53 to baseline levels (Fig. 1i). To check if p53 levels are associated to changes in p53 activity, we measured protein levels of p21, a surrogate marker of p53 activity, and found that PUGNAc increased p21 while OSMI-1 reduced fasting-induced p21 levels (Supplementary Fig. 1e, f).

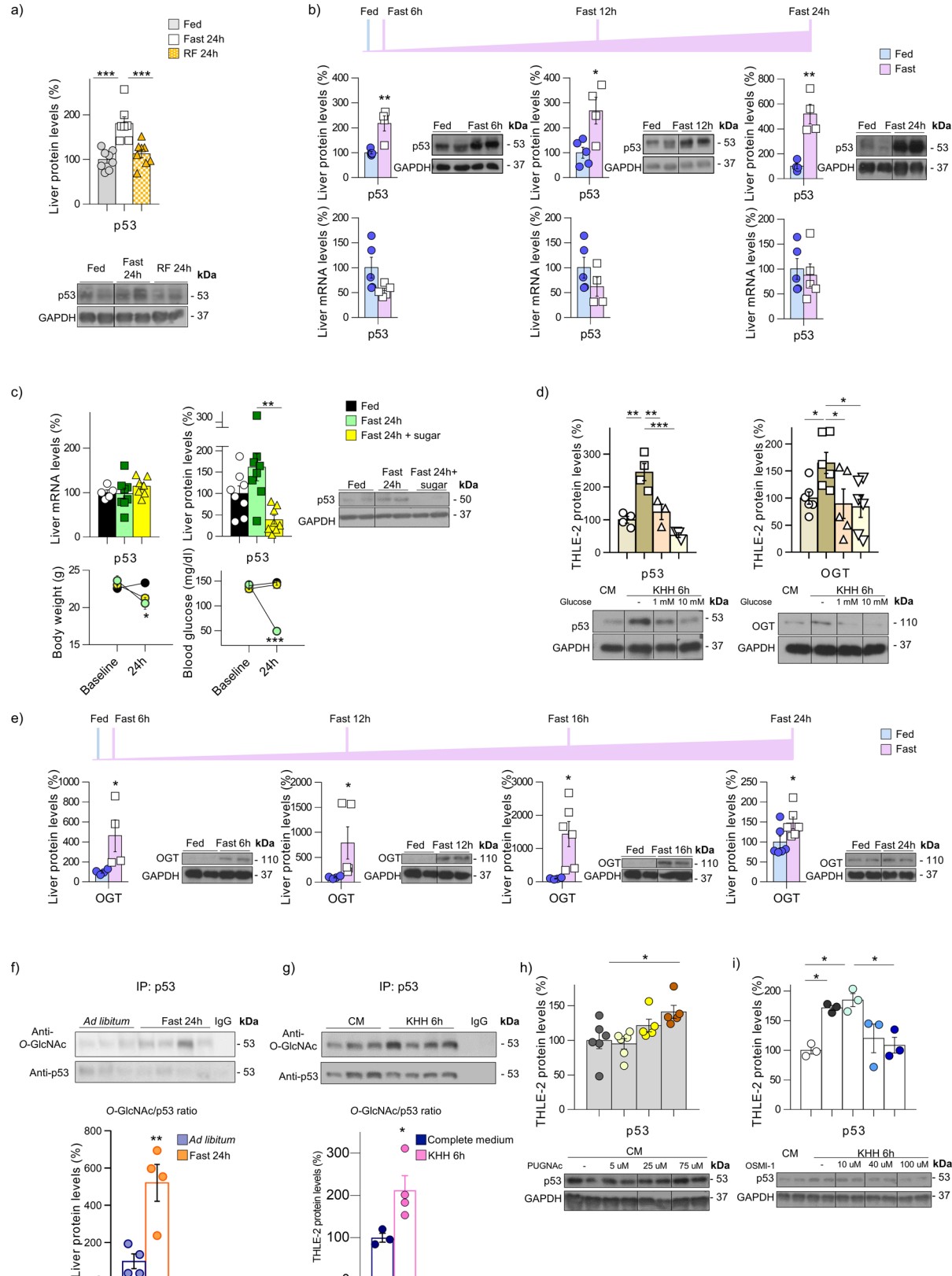

**p53 binds to the PCK1 promoter and regulates its expression**. Once we demonstrated that p53 protein levels are modulated by glucose availability and that this regulation is dependent on O-GlcNAcylation, we studied whether p53 could have an impact on glucose metabolism. Starved cells show upregulated protein levels of p53 and the rate-limiting gluconeogenic phosphoenolpyruvate carboxykinase (PCK1, which converts oxaloacetate into phosphoenolpyruvate and carbon dioxide), without changes in any of the other enzymes involved in glucose synthesis, such as phosphorylated pyruvate dehydrogenase (pPDH), pyruvate carboxylase (PC), or glucose 6-phosphatase (G6Pase) (Fig. 2a). PCK1 protein levels and activity are also upregulated after starvation

**Fig. 1 p53 is stabilized in the liver upon starvation by *O*-GlcNAcylation. a** p53 protein levels in the liver of mice fed ad libitum ($n = 8$), 24-h-fast ($n = 7$) or 24-h-refed (RF) ($n = 8$). **b** p53 protein and mRNA levels in the liver of mice fed ad libitum ($n = 4$ and 5), and mice fasted for 6 h ($n = 4$ and 6), 12 h ($n = 4$), or 24 h ($n = 4$). **c** body weight, blood glucose, p53 protein, and mRNA levels in the liver of mice fed ad libitum ($n = 8$), fasted for 24 h ($n = 8$), and fed with sugar ($n = 8$). **d** p53 and OGT protein levels in THLE-2 cells maintained in complete medium (CM) ($n = 4$ and 6) or starved 6 h in KHH medium without ($n = 4$) or with glucose 1 mM ($n = 3$ and 5) or 10 mM ($n = 3$ and 6). **e** OGT protein levels in the liver of mice fed ad libitum ($n = 4$), and mice fasted for 6 h ($n = 4$), 12 h ($n = 5$), 16 h ($n = 6$), and 24 h ($n = 6$). *O*-GlcNAcylated p53 in: **f** liver of fasted mice ($n = 4$); **g** complete medium (CM) ($n = 3$) and starved cells for 6 h ($n = 4$). **h**) p53 protein levels in THLE-2 cells in the absence ($n = 6$) or presence of PUGNAc 5 ($n = 5$), 25 ($n = 5$), and 75 μM ($n = 5$). **i** p53 protein levels in THLE-2 cells in the absence or presence of OSMI-1 (10, 40, and 100 μM) ($n = 3$). Expression of GAPDH (western blot) and HPRT (qRT-PCR) served as a loading control, and control values were normalized to 100%. Dividing lines indicate splicings within the same gel. Data were presented as mean ± standard error mean (SEM). * denotes $P < 0.05$, ** denotes $P < 0.01$, and *** denotes $P < 0.001$, determined by two-tailed Student's *t*-test (**b**, **e**, **f**, and **g**) or one-way ANOVA followed by Bonferroni post hoc testing (**a**, **c**, **d**, **h**, and **i**). "*n*" denotes independent animals or cell culture wells. Source data are provided as a Source Data file.

and decrease when p53 is silenced (Fig. 2b and Supplementary Fig. 2a, b), while PCK1 activity increases after p53 overexpression (Fig. 2c). The functional relevance of PCK1 as a mediator of p53 action is demonstrated because the increase in glucose levels in the culture medium induced by p53 overexpression, is completely blunted when PCK1 is silenced (Fig. 2d and Supplementary Fig. 2c), while the overexpression of PCK1 rescues glucose production in cells with p53 silenced (Fig. 2e). To corroborate the results obtained with the inhibition of p53 in THLE-2 cells, we used a second cellular model, named Hep3B, which are human hepatocytes that do not express p53. In starved Hep3B cells PCK1 remained unaltered (Fig. 2f) and levels of oxaloacetate were high (Fig. 2g), indicating a decompensated flux because it was not decarboxylated and phosphorylated by PCK1. The rescue of p53 in Hep3B cells (Supplementary Fig. 2d) increased mRNA (Fig. 2h) and protein levels (Fig. 2i) of PCK1. In addition to genetic overexpression of p53, we also treated THLE-2 cells with doxorubicin, a strong activator of p53[28], and found that it induced the levels of p53 and PCK1, but when p53 was silenced, doxorubicin failed to augment PCK1 (Supplementary Fig. 2e).

In mice, hepatic *PCK1* mRNA and protein levels are persistently upregulated after 6, 12, and 24 h of fasting, while PCK1 expression decreases after 2 h of refeeding (Fig. 2j). In line with this, hepatic protein levels of p53 are positively correlated with *PCK1* mRNA and protein levels after 6 h of fasting, and this correlation remains significant after fasting for 12 and 24 h but vanishes upon refeeding (Supplementary Fig. 2f). The positive correlation between p53 and PCK1, detectable already after fasting for 6 h, suggests that p53 could modulate the transcription of PCK1. By searching for potential binding sites of p53 in the PCK1 promoter using the JASPAR database (http://jaspar.genereg.net/), we found three putative motifs for p53 (Fig. 2k). Eight consensus elements for p53 are found in the first −1330 bp upstream from the start transcription site in the PCK1 promoter gene (http://alggen.lsi.upc.es/cgi-bin/promo_v3/promo/promoinit.cgi?dirDB = TF_8.3) (Fig. 2l). To evaluate the possible transcriptional regulation of PKC1 by p53, a luciferase reporter assay in the absence or presence of p53 was carried out using plasmids containing two PCK1 promoter regions, pADTrack-1330-pck1, and pADTrack-490-pck1[29]. Hep3B cells transfected with p53 significantly increase luciferase activity in both PCK1 promoter fragments (Fig. 2l). To confirm the interaction of p53 with the PCK1 promoter, a Chromatin immunoprecipitation (ChIP) assay followed by PCR was performed. The ChIP PCR product significantly increased with the p53 antibody in the p53-transfected AML12 cells using only the most (third region) proximal primers (Fig. 2m, n). Moreover, p21 was significantly increased in the presence of p53 in these cells (Supplementary Fig. 2g).

To further investigate the relevance of *O*-GlcNAc in p53 to modulate PCK1 levels, we treated THLE-2 cells with PUGNAc to increase *O*-GlcNAcylation, and found upregulation in PCK1, while this increase is blunted upon silencing of p53 (Fig. 2o). In agreement with this, the stimulation of *O*-GlcNAc with PUGNAc does not affect PCK1 levels in Hep3B cells, but the rescue of p53 in these cells allows PUGNAc to increase PCK1 (Fig. 2p). Indeed, the ability of PUGNAc to increase the levels of p53 was blunted when p53 was silenced (Supplementary Fig. 2h). The in vitro results demonstrating the direct interaction between p53 and PCK1 promoter were further corroborated in vivo. A ChIP assay was performed in wild-type (WT) mice fed ad libitum or subjected to a 60% calorie restriction, a physiological condition that promotes hepatic gluconeogenesis[30]. Under these conditions, PCK1 promoter bound by p53 under calorie restriction is significantly increased (Fig. 2q) and caloric restriction also increased p21 expression (Supplementary Fig. 2i). Overall, these results indicate that p53 binds to the PCK1 promoter and modulates the transcriptional activity of PCK1 in an *O*-GlcNAc dependent manner.

**O-GlcNAcylation of p53 regulates PCK1 levels**. It was reported that *O*-GlcNAcylation of p53 at Ser149 has a stabilizing effect[20]. To further corroborate the specificity of our pharmacological results using PUGNAc and OSMI-1, we next performed mutagenesis experiments using mutated p53 in Ser149 (p53-S149A). First, we tested whether the overexpression of p53-S149A would result in an increase of protein levels of p53. We tested three different doses (2-4-8 ug) in Hep3B cells, which do not express endogenous p53, and found that p53 levels are upregulated in a dose-dependent manner (Fig. 3a). We transfected Hep3B cells with either p53 WT or p53-S149A and found that p53 WT increased PCK1 protein levels and PCK1 activity, while p53-S149A failed to do so (Fig. 3b, c). Starved Hep3B cells were also transfected with p53-S149A and found that the mutant plasmid was able to increase p53 protein levels, although the expression of PCK1 remained unaltered upon starvation (Fig. 3d). Moreover, PUGNAc failed to modify p53 and PCK1 protein levels in Hep3B cells transfected with p53-S149A (Fig. 3e). We also transfected THLE-2 cells with p53-S149A and found that the mutant plasmid did not increase PCK1 protein levels nor PCK1 activity (Fig. 3f, g). Next, we traced the flux of labeled pyruvate in Hep3B cells using $^{13}$C(1)-pyruvate and monitored by NMR spectroscopy. We transfected these cells with p53 WT and confirm that the gluconeogenic route is overactivated, while the overexpression of mutated p53 (p53-S149A) does not activate the route (Fig. 3h). Overall, these mutagenesis experiments together with our previous pharmacological data, indicate that *O*-GlcNAcylation of p53 in the S149 position is required to activate PCK1. In addition, to understand the relevance of *O*-GlcNAcylated p53 to hepatic gluconeogenesis in vivo, we knocked down OGT in the liver of mice. In another group of mice, p53 was over-expressed after the inhibition of OGT. All the groups were finally placed on fasting

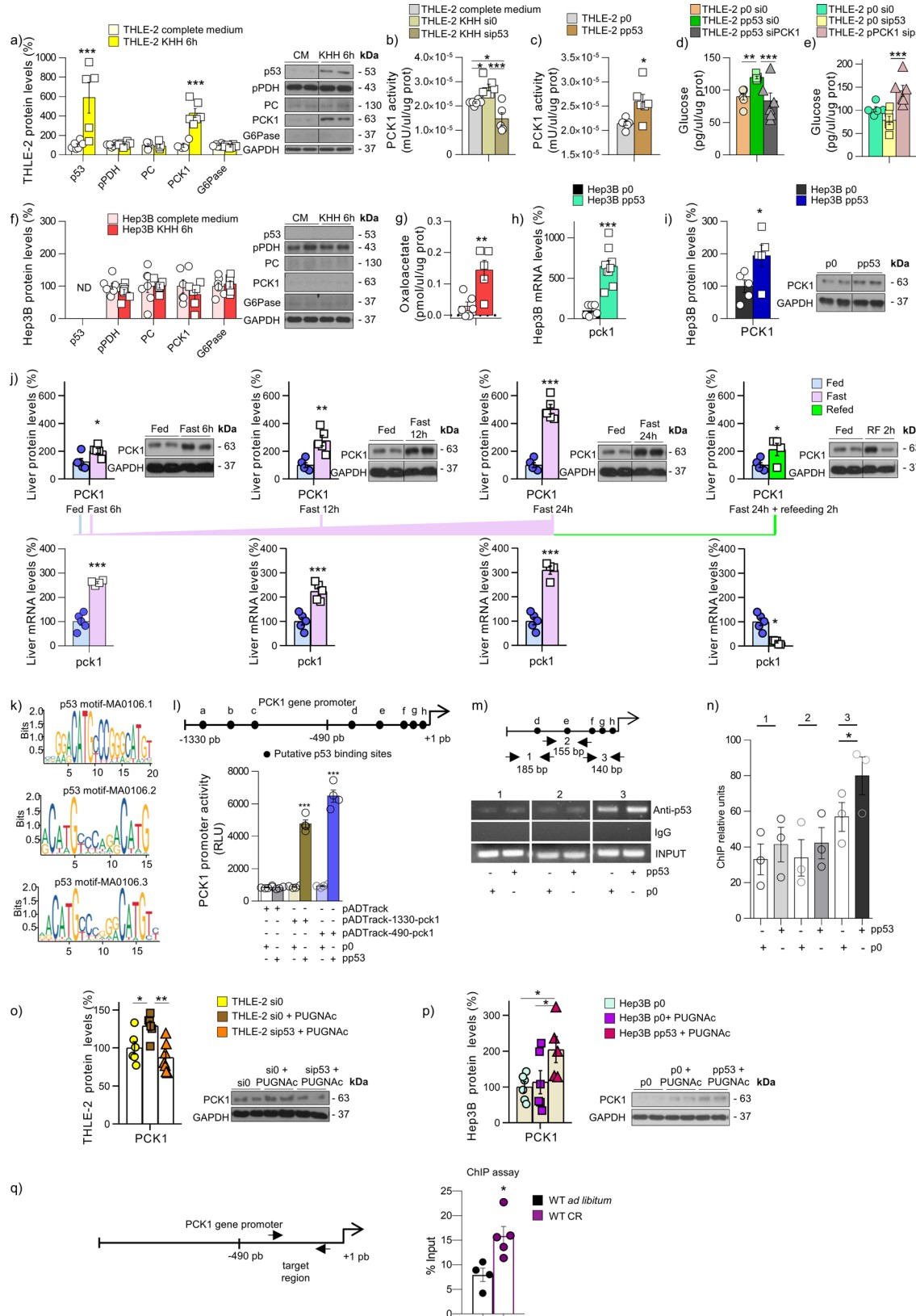

for 24 h (Fig. 3i). The inhibition of OGT downregulates protein levels of p53 and PCK1, as well as blood glucose levels when compared to control, fasted mice (Fig. 3j, k). In mice with a subsequent overexpression of p53 in the liver, we detect that p53 protein levels are increased (Fig. 3k). However, this overexpression of p53 was not able to increase blood glucose levels neither PCK1 expression (Fig. 3j, k). Finally, immunoprecipitated O-GlcNAc is indeed decreased when OGT was knocked down, while the O-GlcNAc/p53 ratio is decreased after the inhibition of OGT, even when p53 is over-expressed (Fig. 3l).

**Fig. 2 p53 binds to PCK1 and regulates its transcriptional activity. a** Protein levels of p53, pPDH, PC, PCK1, and G6Pase ($n = 6$) and **b** PCK1 activity ($n = 6$) in THLE-2 cells in complete medium (CM) or KHH for 6 h. **c** PCK1 activity after the overexpression of p53 ($n = 6$). **d** Glucose levels after the overexpression of p53 and silencing PCK ($n = 6$). **e** Glucose levels after the silencing of p53 and overexpression of PCK1 ($n = 6$). **f** Protein levels of p53, pPDH, PC, PCK1, and G6Pase ($n = 6$) and **g** Oxaloacetate levels in Hep3B cells in complete medium or KHH for 6 h ($n = 6$). **h** mRNA ($n = 7$ and 9) and **i** Protein levels ($n = 5$) of PCK1 after the rescue of p53 in Hep3B cells. **j** PCK1 levels in the liver of mice fed ad libitum ($n = 5$), mice fasted for 6 ($n = 5$), 12 ($n = 5$), or 24-h, and mice 24-h-refed ($n = 5$). **k** Putative motifs for p53 found in the PCK1 promoter. **l** p53 binding sites (black spots) in the PCK1 proximal promoter ($-1330$ bp). Luciferase activity in Hep3B cells transfected as described in the panel ($n = 4$). **m** Diagram of the $-490/+1$ PCK1 gene promoter showing the location of primers used in the ChIP assay. Black spots show the location of the p53 response elements in the PCK1 promoter. Soluble chromatin prepared from AML12 cells transfected with control plasmid or plasmid encoding p53 was immunoprecipitated with an anti-p53 antibody or IgG. An immunoprecipitated proximal sequence of PCK1 was amplified and **n** quantified using three (1, 2, and 3) different pairs of primers. Images are representative of three independent experiments. **o** PCK1 levels in THLE-2 cells and **p** Hep3B cells transfected as indicated in panels and treated with PUGNAc 6 h ($n = 6$). **q** ChIP assay in WT mice fed ad libitum ($n = 4$) or subjected to 60% calorie restriction (CR) ($n = 5$). Diagram of the PCK1 gene promoter showing the location of the amplified region and quantification. Expression of GAPDH (western blot) and HPRT (qRT-PCR) served as a loading control, and control values were normalized to 100%. Dividing lines indicate splicings within the same gel. Data were presented as mean ± standard error mean (SEM). * denotes $P < 0.05$, ** denotes $P < 0.01$, and *** denotes $P < 0.001$, determined by two-tailed Student's $t$-test (**a**, **c**, **f**, **g**, **h**, **i**, **j**, **l**, **n**, and **q**) or one-way ANOVA followed by Bonferroni post hoc testing (**b**, **d**, **e**, **o**, and **p**). "$n$" denotes independent animals or cell culture wells. Source data are provided as a Source Data file.

---

**Mice lacking p53 in the liver show impaired gluconeogenesis**. Since our data show direct regulation of PCK1 by p53, we generated two p53 knockout animal models to study in detail the role of endogenous hepatic p53 in glucose metabolism. One model was generated by injecting AAV8-Cre into the tail vein of adult p53 floxed mice (Fig. 4a and Supplementary Fig. 3a). In the other model, p53 floxed mice were crossed with mice which express Cre recombinase under the control of the albumin gene promoter with α–fetoprotein enhancer (Alfp-Cre), resulting in the lack of p53 specifically in the liver (p53 LKO) (Fig. 4b and Supplementary Fig. 4a). While none of these models show differences in food intake or body weight (Supplementary Figs. 3b–4b) nor in glucose tolerance or insulin sensitivity (Supplementary Figs. 3b–4c), they show lower glucose levels after the administration of pyruvate (Fig. 4a, b) indicating impaired gluconeogenesis. Of note, the depletion of p53 does not affect hepatic glucose production (HGP) with glycerol (which bypasses the requirement of PCK1 expression) (Supplementary Figs. 3d–4d). Importantly, the effects of p53 on glucose homeostasis are not caused by alterations in apoptosis and/or cell proliferation, since immunostainings for cleaved caspase 3 and ki67 in mice lacking hepatic p53 do not show differences when compared to control mice (Supplementary Figs. 3e–4e). These effects are, however, clearly visible in the liver with hepatocellular carcinoma as expected.

We next restricted chow-fed mice to 40% of their daily food intake[30] resulting in 60% calorie restriction, a condition that increases liver gluconeogenesis. While both models of genetic depletion of hepatic p53 show similar weight evolution and blood glucose levels when fed ad libitum (Supplementary Figs. 3f–4f), during caloric restriction they are unable to maintain blood glucose at the same level as control mice (Fig. 4c, d). Consistent with this, p53 LKO mice have lower HGP (Fig. 4e). We then rescue p53 in the liver of the knockout mice. In concordance with a key role of p53 in glucose production, the rescue of p53 in the liver of both p53-depleted animal models restore blood glucose after caloric restriction to the levels found in control mice (Fig. 4f, g). The changes in hepatic PCK1 levels are in agreement with the circulating glucose profiles; in control mice under caloric restriction, glucose levels maintain stable and hepatic PCK1 protein levels are upregulated, whereas in both models of hepatic p53 depletion both glucose and PCK1 levels decrease under caloric restriction. The rescue of p53 reverses PCK1 to similar levels as in control mice (Fig. 4h, i). Following the caloric restriction, mice were supplied with their daily amount of food during calorie restriction, which results in a prompt rise in blood glucose (Supplementary Figs. 3g–4g). No differences are detected

between control mice and p53-depleted mice with respect to serum NEFAs and ketone bodies (Supplementary Figs. 3h–4h). The two animal models of hepatic deletion of p53 and further rescue of p53 in mice under caloric restriction did not alter apoptosis and/or cell proliferation, as immunostainings for cleaved caspase 3 and ki67 did not show differences when compared to control mice (Supplementary Fig. 5).

Next, we performed a gain-of-function experiment, where p53 was specifically upregulated in the liver of Alfp-Cre mice using an AAV8-DIO-p53-EGFP, which Cre-dependently overexpresses p53. Our results indicate that while these mice show no differences in body weight, food intake, glucose tolerance, or insulin sensitivity (Supplementary Fig. 6a, b), they have higher circulating glucose after a pyruvate tolerance test (Fig. 4j). Moreover, when subjected to 60% calorie restriction, mice over-expressing p53 in the liver show higher blood glucose levels than control mice (Fig. 4k), but without differences when fed ad libitum conditions (Supplementary Fig. 6c) or after a meal (Supplementary Fig. 6d). In addition, the overexpression of p53 did not affect protein levels of cleaved caspases 3 and 7 or immunostainings of cleaved caspase 3 and ki67 when compared to control mice (Supplementary Fig. 7). Overall, these data support a major role for p53 in hepatic gluconeogenesis.

**Glucagon and forskolin require p53 to stimulate PCK1 and raise glucose levels.** Once we demonstrated that endogenous p53 plays a physiological role in gluconeogenesis under conditions of caloric restriction, we next aimed to investigate whether p53 could also mediate the effect of glucose counterregulatory hormones. Glucagon, which is stimulated by low plasma glucose levels, increases HGP. Thus, we treated THLE-2 cells with glucagon and found a significant increase in p53 protein levels (Fig. 5a). This glucagon-induced p53 stabilization is abolished by the pretreatment of cells with OSMI-1 (Fig. 5b), indicating that the effect of glucagon on p53 is mediated by $O$-GlcNAc. Next, p53 was silenced in cells incubated with starvation medium or glucagon (Fig. 5c). Under these conditions, although glucagon induces the phosphorylation of CREB (pCREB), it fails to increase PCK1 levels (Fig. 5c) and PCK1 activity (Fig. 5d). Therefore, these data suggest that p53 is downstream of pCREB in the glucagon signaling pathway to modulate PCK1. To corroborate this, we treated cells with forskolin, which, similar to the effect of glucagon, raises intracellular cAMP levels via activation of PKA. Forskolin triggers pCREB (a surrogate marker of PKA activity) and increases p53 (Fig. 5e). However, after silencing p53, forskolin is unable to increase PCK1 (Fig. 5f) and glucose levels

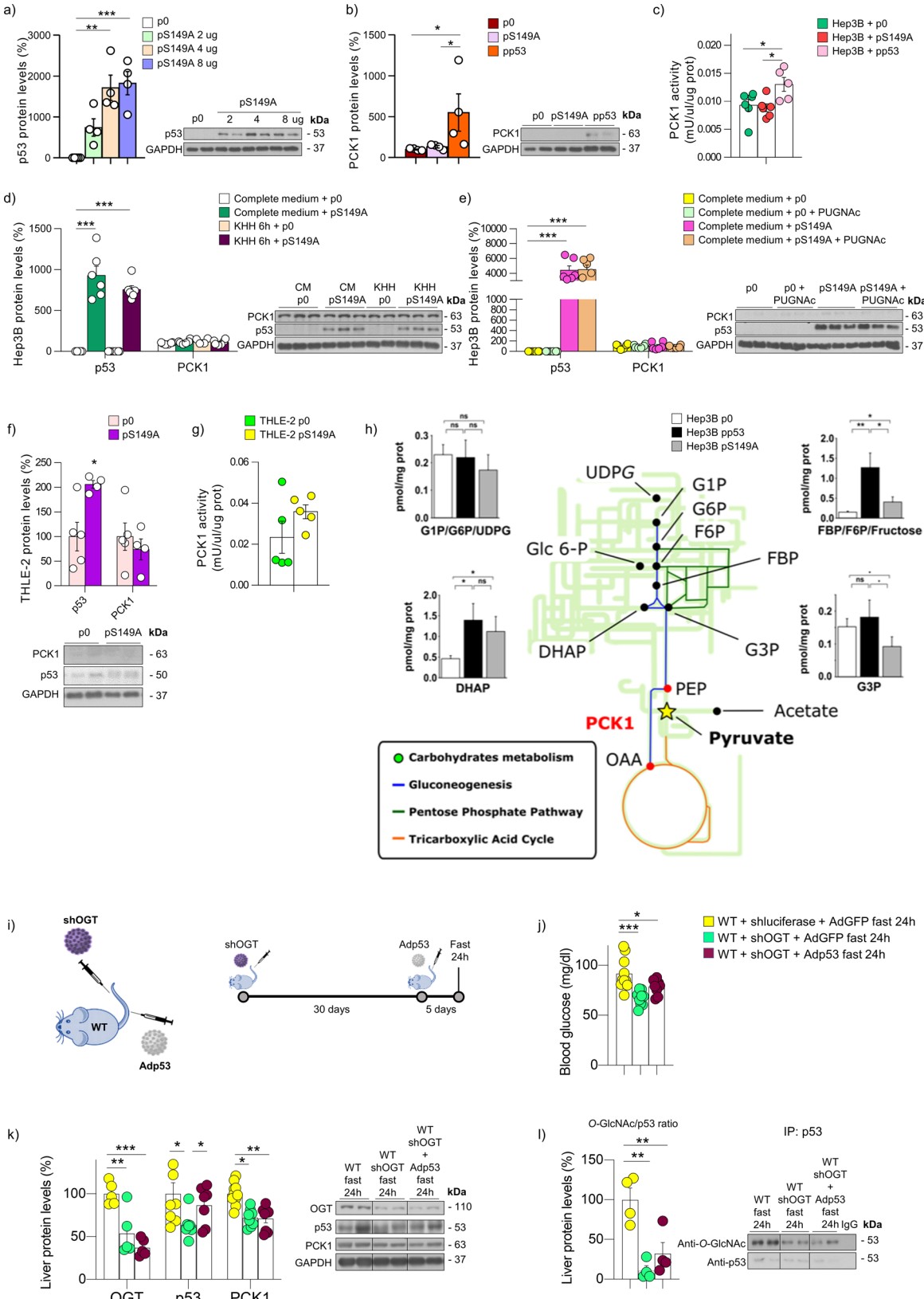

(Fig. 5g). Consistent with this, both glucagon (Fig. 5h) and forskolin (Fig. 5i) do not upregulate PCK1 in Hep3B cells that lack p53. However, when p53 is rescued in Hep3B cells, the capacity of forskolin to induce PCK1 levels (Fig. 5j) and to increase glucose levels in the medium is restored (Fig. 5k). In order to address the relevance of *O*-GlcNAcylation of p53 in PKA-pCREB pathway

induced gluconeogenesis (a common pathway between glucagon and adrenaline), we carried out p53-S149A mutagenesis in Hep3B treated with forskolin and found that, while it induced increases of pCREB and p53, this compound was not able to upregulate PCK1 protein levels when p53 was mutated (Supplementary Fig. 8a).

**Fig. 3 O-GlcNAcylation of p53 regulates PCK1 protein levels. a** p53 protein levels in Hep3B cells transfected with empty plasmid (p0), or a plasmid encoding p53-S149A mutant (2-4-8 µg) ($n = 4$). **b** PCK1 protein levels ($n = 4$) and **c** PCK1 activity ($n = 6$) in Hep3B cells transfected with empty plasmid, a plasmid encoding p53-WT (pp53) or a plasmid encoding p53-S149A (pS149A). **d** p53 and PCK1 protein levels in Hep3B transfected with empty plasmid or a plasmid encoding p53-S149A maintained in complete medium (CM) or KHH medium for 6 h ($n = 6$). **e** p53 and PCK1 protein levels in Hep3B transfected with empty plasmid or a plasmid encoding p53-S149A and treated with vehicle or PUGNAc 75 µM ($n = 6$). **f** p53 and PCK1 protein levels (p0 $n = 5$ and pS149A $n = 4$) and **g** PCK1 activity ($n = 5$) in THLE-2 cells transfected with empty plasmid or a plasmid encoding p53-S149A. **h** Metabolomic flux in Hep3B cells transfected with empty plasmid, a plasmid encoding p53-WT or a plasmid encoding p53-S149A ($n = 3$). **i** Schematic representation of the animal model and procedure: inhibition of OGT with a lentivirus encoding shOGT and overexpression of p53 with an adenovirus encoding p53, in the liver of WT mice finally placed on fasting for 24 h. **j** Blood glucose levels ($n = 10$) and **k** protein levels of OGT, p53, and PCK1 in the liver of each of the groups described above ($n = 5$ and 8). **l** Immunoprecipitated p53 and O-GlcNAc/p53 ratio ($n = 4$). Expression of GAPDH served as a loading control, and control values were normalized to 100%. Dividing lines indicate splicings within the same gel. Data were presented as mean ± standard error mean (SEM). * denotes $P < 0.05$, ** denotes $P < 0.01$, and *** denotes $P < 0.001$, determined by two-tailed Student's $t$-test (**f**) or one-way ANOVA followed by Bonferroni post hoc testing (**a, b, c, d, e, h, j, k**, and **l**). "$n$" denotes independent animals or cell culture wells. Source data are provided as a Source Data file.

Notably, to further investigate the physiological relevance of this in vitro scenario, mice with virogenetic ablation of hepatic p53 and p53 LKO mice were treated with glucagon. Our data show that glucagon upregulates protein levels of p53 (Supplementary Fig. 8b), an effect that was not reproduced in glucagon receptor knockout mice (Supplementary Fig. 8c). Glucagon fails to boost blood glucose in the two animal models lacking hepatic p53 (Fig. 5l, m). Of note, glucagon increases the O-GlcNAcylation of p53 (Fig. 5n) and OGT protein levels in WT but not in glucagon receptor-deficient mice, and promotes the physical interaction between OGT and p53 (Supplementary Fig. 8b–d). Overall, our results indicate that both glucagon and forskolin require p53 to stimulate PCK1, ultimately increasing glucose production (Fig. 5o).

**p53 modulates the gluconeogenic capacity of adrenaline and glucocorticoids.** In addition to glucagon, adrenaline and glucocorticoids are key counterregulatory hormones, although their mechanism of action and time course are slightly different. Similar to glucagon, adrenaline stimulates hepatic gluconeogenesis and reduces glucose uptake by the muscle. Glucocorticoids, in addition to impair insulin secretion, induce hyperglycemia by increasing the synthesis of gluconeogenic enzymes and reducing peripheral tissue-glucose uptake by increasing insulin resistance. In contrast to glucagon and adrenaline, the onset of the effects of glucocorticoids on glucose occurred at later times and is longer-acting. Adrenaline- and hydrocortisone-treated cells show increased protein levels of p53 in a dose-dependent manner in vitro (Fig. 6a, h). This adrenaline- and glucocorticoid-induced p53 stabilization is abolished by the pretreatment of cells with OSMI-1 (Fig. 6b, i), indicating that the effects of both adrenaline and glucocorticoids on p53 are mediated by O-GlcNAc. These two hormones also stimulate PCK1 levels and PCK1 activity but fail to exert these actions when p53 is silenced (Fig. 6c, j, d and k). In agreement with this, they are also unable to upregulate PCK1 in Hep3B cells (Fig. 6e, l). The physiological relevance of this data is validated by our in vivo findings. Mice treated with adrenaline and glucocorticoids show increased protein levels of p53 and PCK1 in the liver (Supplementary Fig. 8e–g). Adrenaline augments OGT while glucocorticoids decrease OGA (Supplementary Fig. 8f–h). Glucocorticoids were not able to increase PCK1 levels in cells transfected with p53-S149A (Supplementary Fig. 8i), but did so after being transfected with p53 WT (Supplementary Fig. 8j). Both adrenaline- and glucocorticoid-induced hyperglycemia in control mice, does not occur in animal models with depletion of p53 (either by virogenetic ablation of hepatic p53 or p53 LKO) (Fig. 6f, g, m and n). Taking together, these data indicate that O-GlcNAcylation of p53 is required for the actions of glucagon, adrenaline, and glucocorticoids.

**Overexpression of p53 in hepatocytes reduces insulin sensitivity.** Our findings indicating that p53 plays a relevant role in mediating the gluconeogenic effect of the most important glucose counterregulatory hormones, prompted us to further evaluate the potential implication of p53 in the actions of the glucose-lowering hormone insulin. Insulin stimulates glucose uptake in muscle and in adipose tissue as well as inhibiting glucose production in the liver[31], and decreased insulin secretion is considered the first line of defense against hypoglycemia. In starved cells, insulin decreases p53 protein levels in a dose-dependent manner (Fig. 7a) and restores basal levels of p53 and PCK1 (Fig. 7b). When p53 is over-expressed in hepatocytes, insulin loses its capacity to inhibit PCK1 protein levels and PCK1 activity (Fig. 7c, d) and to reduce glucose levels (Fig. 7e). In addition, when p53 levels are induced by PUGNAc, insulin fails to downregulate PCK1 (Fig. 7f). Since AKT is a serine/threonine-specific protein kinase that plays a key role in the insulin-signaling pathway, we treated THLE-2 cells with the AKT activator SC79, which increased pAKT while decreases p53 and PCK1 (Fig. 7g), but the overexpression of p53 blocks the ability of SC79 to inhibit PCK1 (Fig. 7h).

In line with the in vitro data, we observe increased PCK1 levels when p53 is over-expressed in the liver of WT mice (Fig. 7i). As expected, insulin increases pAKT and decreases p53 and PCK1 in the liver of WT mice (Fig. 7j). However, in mice with hepatic p53 overexpression, insulin does not inhibit PCK1, despite high pAKT levels (Fig. 7j). We then used Alfp-Cre mice and injected AAV-DIO-EGFP or AAV-DIO-p53-EGFP, as a second animal model over-expressing p53 specifically in hepatocytes, and injected insulin in both groups. In control mice, insulin causes the expected marked decrease in blood glucose levels after 20 min (Fig. 7k). However, this hypoglycemic effect as well as the inhibition of PCK1 are blunted in mice over-expressing p53 in the liver (Fig. 7k, l). Altogether, these findings suggest a negative association between insulin-dependent activation of AKT and p53 in the regulation of PCK1 and blood glucose levels (Fig. 7m).

**Hepatic p53 affects postprandial hyperglycemia and is associated with human T2D.** Given the relevance of p53 for regulating insulin sensitivity, we further assessed whether hepatic p53 could also play a role in regulating high-fat diet (HFD)- and postprandial-induced hyperglycemia. First, we measured p53 and OGT protein levels in the liver of WT mice under HFD for 4 days, a model that presents an early insulin resistance without differences in body weight and fatty liver phenotype, and found a significant increase in both of them (Fig. 8a). However, when the two animal models deficient for p53 in the liver were challenged to the same diet and time, p53 and OGT remained stable (Fig. 8b, c). Next, we induced p53 expression in the liver of Alfp-Cre mice fed a standard diet and allowed them to eat ad libitum following an overnight fast. Both control and mice over-expressing p53 eat

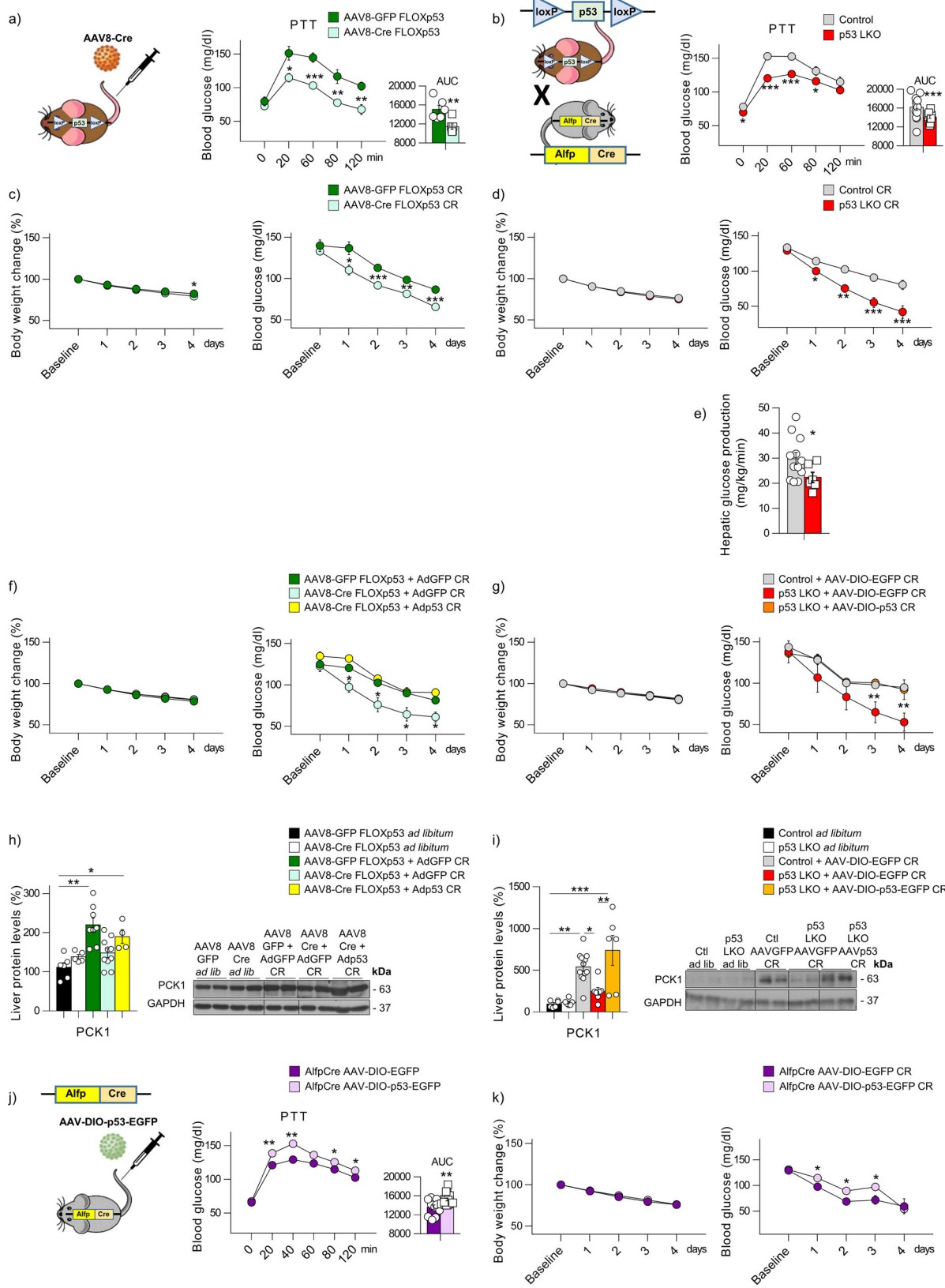

the same amount of food within 4 h (Supplementary Fig. 9a), but blood glucose is significantly higher in mice over-expressing p53 compared to control mice (Fig. 8d). The same result is found when mice receive an HFD (Fig. 8e and Supplementary Fig. 9b). Next, we performed similar experiments in both mouse models of hepatic p53 depletion. Mice with virogenetic knockdown of p53

in the liver show reduced postprandial glucose levels, both when they have free access to standard food and HFD (Fig. 8f, g and Supplementary Fig. 9c, d). In line with this, p53 LKO are also protected against postprandial hyperglycemia independent of the amount and type of diet consumed during 4 h after overnight fasting (Fig. 8h, i and Supplementary Fig. 9e, f). It is important to

**Fig. 4 Mice lacking p53 in the liver do not maintain glucose levels during caloric restriction. a** Pyruvate tolerance test (PTT) in p53floxed mice injected with either AAV8-GFP (control group; $n = 6$) or AAV8-Cre ($n = 7$). **b** PTT in control mice ($n = 13$) and p53 LKO mice ($n = 12$). Body weight and blood glucose levels in: **c** p53floxed mice injected with either AAV8-GFP ($n = 10$) or AAV8-Cre ($n = 14$) and **d** p53 LKO mice fed ad libitum ($n = 12$) or subjected to 60% calorie restriction (CR) for 4 days ($n = 17$). **e** Hepatic glucose production in both p53 LKO ($n = 6$) mice and their control littermates ($n = 12$) on the fourth day of calorie restriction. Body weight and blood glucose levels after the rescue of hepatic p53 in: **f** p53 floxed mice injected with either AAV8-GFP or AAV8-Cre ($n = 6$ and 7 per group) and **g** p53 LKO mice, subjected to 60% calorie restriction for 4 days ($n = 10$, 5, and 7 per group). **h, i** Hepatic PCK1 protein levels in each of the groups described above. **j** PTT in mice over-expressing GFP ($n = 12$) or p53 in the liver ($n = 15$ per group). **k** Body weight and blood glucose levels in mice over-expressing p53 in the liver after 60% calorie restriction for 4 days. The area under the curve (AUC) is provided. Expression of GAPDH served as a loading control, and control values were normalized to 100%. Dividing lines indicate splicings within the same gel. Data were presented as mean ± standard error mean (SEM). * denotes $P < 0.05$, ** denotes $P < 0.01$, and *** denotes $P < 0.001$, determined by two-tailed Student's $t$-test (**a, b, c, d, e, j,** and **k**) or one-way ANOVA followed by Bonferroni post hoc testing (**f, g, h,** and **i**). "$n$" denotes independent animals. Source data are provided as a Source Data file.

highlight that the effects of the lack and overexpression of p53 on glucose were maintained at long-term, since mice lacking or over-expressing p53 in the liver were monitored after 4, 5, and 6 months and the pyruvate tolerance test showed the same results as in young mice (Supplementary Fig. 10). Finally, we examined p53 protein levels in the liver of a diabetic animal model like mice lacking leptin receptor (*db/db*) and found that p53 and PCK1 are significantly elevated (Fig. 8j).

Excessive postprandial hyperglycemia is a feature developed by many patients with T2D[32]. Thus, we explored the potential clinical relevance of hepatic p53 in human diabetes. For this, we assessed p53 levels in the liver of patients with obesity further subclassified according to their normoglycemia (NG) or T2D (Supplementary Table 1). While *p53* mRNA expression remains unaltered between the two groups (Fig. 9a), protein levels of p53 are significantly higher in the liver of patients with T2D, which is concomitant with increased PCK1 levels (Fig. 9b). In line with this, p53 protein levels, but not *p53* mRNA, are positively correlated with glycemia 2 h after an oral glucose tolerance test (OGTT) (Fig. 9c) and with insulin resistance, as evidenced by the homeostasis model assessment (HOMA) index (Supplementary Table 1). Multiple linear regression analysis reveals that insulin resistance, together with the age, is a major determinant of hepatic p53 protein levels but not *p53* mRNA (Supplementary Table 2).

The expression of OGT and glutamine-fructose-6-phosphate amidotransferase (GFAT), the first and rate-limiting enzyme in the synthesis of UDP-GlcNAc, are significantly increased in the liver of T2D patients, indicating higher levels of *O*-GlcNAc (Fig. 9d). Hepatic *GFAT1* and *GFAT2* mRNA levels are positively correlated with the glycemia 2 h after an OGTT (Fig. 9e). Moreover, the univariate analyses reveal a strong association between the surrogate index of insulin resistance, HOMA, and hepatic *GFAT2* transcript levels (Supplementary Table 3). Accordingly, multivariate analysis show that HOMA-IR contributed independently to 27.9% ($P < 0.05$) of hepatic *GFAT2* gene expression variance after controlling for the effects of age, sex, and BMI (Supplementary Table 4). To note, neither p53 nor PCK1 were significantly associated with the NAS score (Supplementary Tables 5 and 6), indicating that the results are not affected by liver damage.

## Discussion
p53 has been the most studied gene in cancer research over the last 30 years since mutations in Trp53 are highly common, although quite variable, reaching more than 90% in some instances. From data gleaned in the past, p53 has emerged as a central hub in a molecular network controlling cellular redox homeostasis, cell proliferation, and death. The interest in the study of p53 is magnified by the fact that, unlike other tumor suppressor genes, usually associated with inactivating mutations,

genetic alterations in p53 could lead to both inactivation or gain-of function activities. In addition to the classical view of p53, noncanonical functions of p53, like its role in glucose home-ostasis, are also emerging as elements that may play a key role in cell metabolism and tumor suppression[3,10,33,34]. Importantly, some features of the glucose metabolism deregulation that occur in several types of cancers (i.e., modified glycolysis) are similar to those found in T2D. The role of p53 in glucose metabolism has previously been studied in the complex scenario of a tumoral cell. Under these conditions of severe stress, one of the strategies used by p53 is the reduction of glucose 6 phosphatase dehydrogenase[7] and PCK1[35] to limit glucose availability and promote the death of the tumor cell. However, the contribution of p53 in the control of glucose homeostasis in physiological situations remains largely unsolved. In agreement with a previous study[12], we find that hepatic p53 is dynamically regulated by nutrient availability, being stabilized upon starvation. We show that the stabilization of p53 during starvation is dependent of *O*-GlcNAc. This post-transcriptional modification is known to prolong the half-life of p53 after DNA damage[20]. The stabilization of p53 in conditions of food deprivation, which is stressful to the cell, is consistent with the role of p53 as a stress response factor. In this work, we demonstrate that *O*-GlcNAcylation of p53 in the liver is essential for the endogenous HGP. When p53 is mutated at Serine 149, where *O*-GlcNAcylation of p53 has a stabilizing effect[20], it lacks its ability to regulate PCK1 levels. It has been reported that *O*-GlcNAcylation also occurs in several molecules that induce glu-coneogenesis, such as CRTC2[36], FOXO1[37], or PGC1α[38], in the complex context of hyperglycemia and diabetes, where the hex-osamine biosynthetic pathway (HBP), the canonical source of GlcNAc, is overstimulated. Under these conditions, *O*-GlcNA-cylation of CRTC2, FOXO1, and PGC1α prevents their degra-dation, promoting their half-life, which results in the increased expression of gluconeogenic genes. Interestingly, *O*-GlcNAcyla-tion has not only been reported under hyperglycemic conditions but also upon nutrient deprivation[17,23–25,39]. As HBP flux is decreased during fasting, it has been suggested that one potential mechanism that could mediate this effect is the called "salvage pathway", by which some proteins lose its *O*-GlcNAc moiety by the OGA activity, and free GlcNAc can be recycled and attached to new target proteins. Another potential mechanism that could explain the increase in *O*-GlcNAcylation during fasting is an upregulation of OGT expression, which may raise cellular *O*-GlcNAcylation levels despite the reduced intracellular availability of UDP-GlcNAc[17,23–25,39]. In the present study, we described that OGT protein levels are upregulated during fasting, in both in vitro and in vivo models, as well as after the administration of counterregulatory hormones, which can explain the *O*-GlcNA-cylation of p53 found under these conditions. Moreover, as it happens to CRTC2, FOXO1, and PGC1α, we found that p53 is also upregulated in the liver of diabetic patients, highlighting the

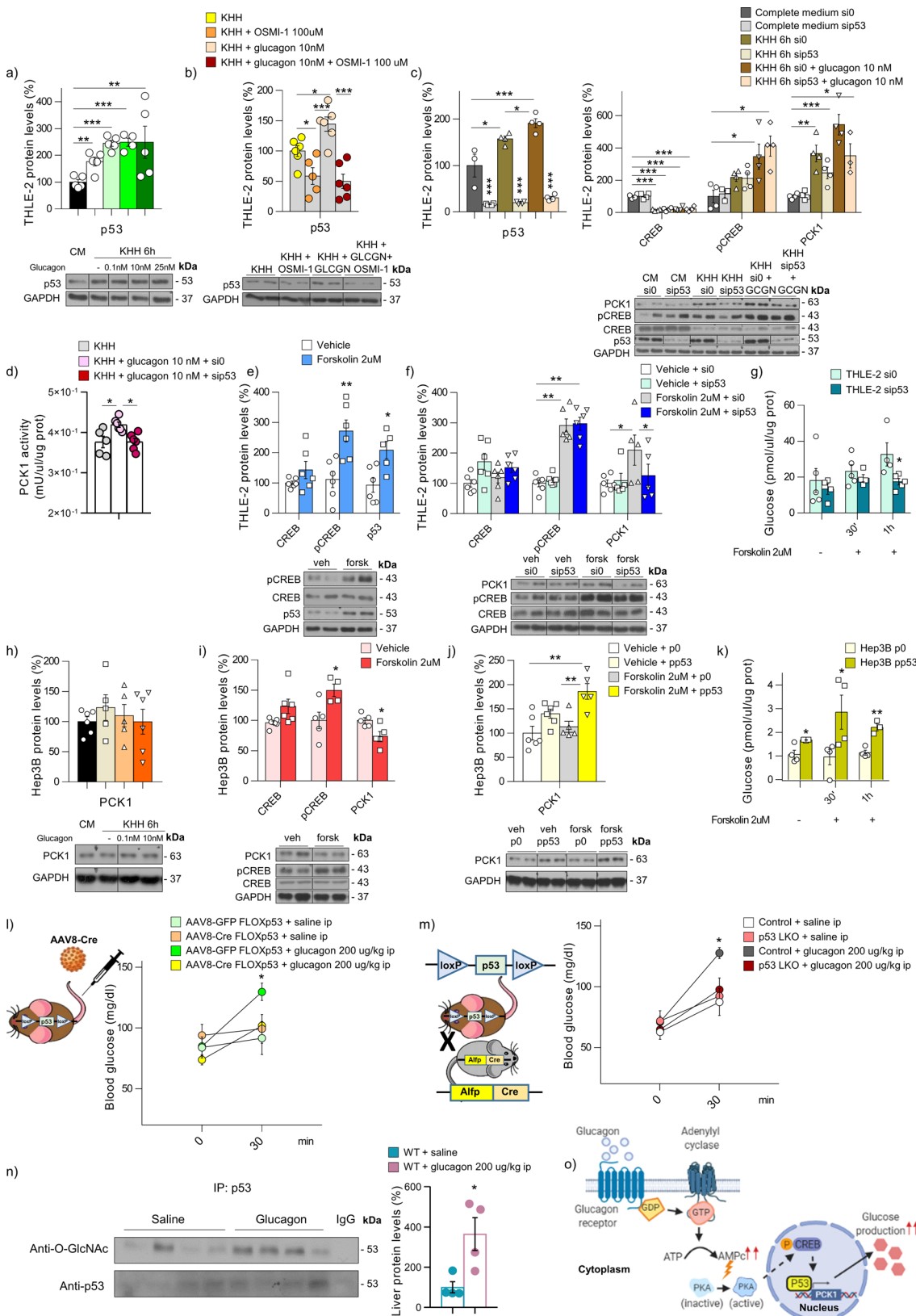

relevance of *O*-GlcNAcylation as a shared and common mechanism among them. Although further studies are needed to delineate this aspect, the fact that transcriptional mediators of glucose production are *O*-GlcNAcylated suggest that this post-translational modification plays an important role to defend the organism from energy deficiency during nutrient deprivation.

Indeed, p53 protein levels are positively correlated with PCK1 in both mice and human hepatocytes deprived of nutrients. These results are in line with a previous study showing that cells treated with nutlin-3a, a p53-activating agent, showed an upregulation in the expression of some gluconeogenic genes[11]. Moreover, our present findings demonstrate that p53 binds to

**Fig. 5 Hepatic p53 mediates the gluconeogenic action of glucagon.** p53 levels in **a** THLE-2 cells treated with glucagon ($n = 5$); **b** THLE-2 cells treated with glucagon and OSMI-1 ($n = 6$). **c** Protein levels of p53, CREB, pCREB, and PCK1 ($n = 4$) and **d** PCK1 activity ($n = 6$) in THLE-2 cells treated with empty-siRNA or siRNA-p53, and then kept in KHH or KHH + glucagon. **e** Protein levels of p53, CREB, and pCREB in THLE-2 cells treated with forskolin 2 μM ($n = 6$). **f** Protein levels of CREB, pCREB, and PCK1 in THLE-2 cells treated with empty-siRNA or siRNA-p53, and then challenged with forskolin ($n = 6$). **g** Glucose levels in THLE-2 cells treated with empty-siRNA or siRNA-p53 incubated with or without forskolin ($n = 5$). **h** PCK1 levels in Hep3B starved cells in the presence or absence of glucagon ($n = 6$). **i** Protein levels of CREB, pCREB, and PCK1 in Hep3B cells treated with forskolin ($n = 6$). **j** Protein levels of PCK1 and **k** glucose levels in Hep3B cells transfected with empty plasmid or plasmid encoding p53 and then treated with forskolin ($n = 6$). **l** Blood glucose levels in p53 floxed mice injected with AAV8 expressing either GFP ($n = 6$ and 7) or Cre ($n = 7$ and 9) and then treated with saline or glucagon (200 μg kg$^{-1}$). **m** Blood glucose levels in p53 LKO mice and their control littermates treated with saline or glucagon ($n = 6$). **n** O-GlcNAcylation of p53 in WT mice treated with saline or glucagon (200 μg kg$^{-1}$) ($n = 4$). **o** Schematic representation of the role of p53 in glucagon-induced gluconeogenesis. Expression of GAPDH served as a loading control, and control values were normalized to 100%. Data were presented as mean ± standard error mean (SEM). * denotes $P < 0.05$, ** denotes $P < 0.01$, and *** denotes $P < 0.001$, determined by two-tailed Student's $t$-test (**e**, **g**, **i**, **k**, and **n**) or one-way ANOVA followed by Bonferroni post hoc testing (**a**, **b**, **c**, **d**, **f**, **j**, **l**, and **m**). "$n$" denotes independent animals or cell culture wells. Source data are provided as a Source Data file.

the PCK1 promoter, and thereby increases PCK1 transcriptional activity. The direct interaction between p53 and PCK1 is functionally translated to its ability to activate gluconeogenesis, as demonstrated by the lower capacity of mouse models lacking p53 in the liver to produce glucose from pyruvate. Thus, specific depletion of hepatic p53 reproduces the findings observed in p53-null mice, which also show lower blood glucose after pyruvate injection[13]. In line with this, mice lacking p53 in the liver are unable to maintain blood glucose levels during several days of calorie restriction, which in WT mice relies on the induced expression of gluconeogenic genes[40]. Notably, several conditional KO mice had to receive glucose since they had glucose levels lower than 20 mg/dL and their life was at risk without this supplement. This is somehow in line with the lower glucose levels upon starvation in mice lacking p53 in the liver[12], but our data indicate that hepatic p53 is essential for survival in long-term situations of low nutrient availability. The ability of p53 to maintain glucose levels seems to depend on its capacity to regulate PCK1, because PCK1 expression is lower in calorie-restricted mice without hepatic p53. Consistent with this, the recovery of p53 in the liver of mice where the gene was previously depleted, restores PCK1 protein levels and blood glucose, and the silencing of PCK1 in hepatocytes blunts the p53-induced glucose production.

When food is not accessible, glucose homeostasis is tightly regulated by a number of hormones and physiological responses. The initial response that prevents a decline in blood glucose concentration is the drop of insulin secretion[41,42], followed by secretion of counterregulatory hormones as glucose levels fall below the optimal physiologic range[41,42]. Strikingly, we find that p53 is required for all the studied counterregulatory hormones implicated in glucose homeostasis. The glucagon-stimulated transcription of PCK1 is mediated by cAMP signaling in hepatocytes[43]. Our in vivo and in vitro findings demonstrate that both glucagon and the cAMP pathway activator forskolin, increase p53 protein levels and require the presence of p53 in hepatocytes to exert their gluconeogenic action. In vitro, the silencing of p53 in hepatocytes incubated with glucagon and forskolin prevents the expected increase in PCK1 or glucose levels in the medium. In parallel, animal models of hepatic p53 depletion do not increase blood glucose concentration in response to glucagon. Identical results are obtained for adrenaline that, similarly to glucagon, also raises cAMP levels to induce hepatic glucose production[44], and for glucocorticoids, which do not increase cAMP levels but uses other pathways to upregulate PCK1 and glucose production[45]. Therefore, the presence and binding of p53 to PCK1 promoter seems to be essential for gluconeogenic hormones exerting their effect via PCK1. Importantly, these hormones require the O-GlcNAcylation of p53 because the mutation of p53 in Serine 149 blunts their gluconeogenic action.

PCK1 is not just necessary for the function of counterregulatory hormones but also for the capacity of insulin to lower glucose production. Insulin inhibits HGP by directly diminishing mRNA expression of PCK1[46], and transgenic mice over-expressing PCK1 have insulin resistance[47] and hyperglycemia[48]. Our data demonstrate that in normal conditions, insulin decreases p53 protein levels in the liver. However, it fails to decrease PCK1 and glucose concentration when p53 is over-expressed in hepatocytes or in the liver of mice. The downregulation of p53 levels by insulin seem to be mediated by Akt. It has been reported that Akt mediates the negative control of p53 levels through enhancing MDM2 (murine double minute 2)-mediated targeting of p53 for degradation[49]. This suggests that the degradation of p53 plays a pivotal role in the inhibition of glucose production by insulin and that p53 is involved in both up-and down-regulation of gluconeogenesis.

Considering that our preclinical results in animal models and human hepatic cell lines unequivocally demonstrate that gluconeogenic hormones require p53 and that overexpression of p53 causes insulin resistance, we next evaluated the translational value of these findings, in light of the larger contribution of hepatic gluconeogenesis to HGP in small animals as compared to humans. Previous reports have shown that in addition to the well-established role of p53 in cancer, p53 mutations are also related with metabolic alterations in humans. Patients with the Li Fraumeni syndrome, which is an autosomal dominant cancer predisposition disorder caused by some p53 mutations, have increased oxidative phosphorylation of skeletal muscle[50]. Moreover, the protein encoded by the gene mutated in ataxia-telangiectasia patients, named Atm, phosphorylates p53 and these patients show increased cholesterol levels and insulin resistance[51], while the expression of p53 in human white adipose tissue is dually affected by inflammation and insulin resistance[52]. Our results demonstrate that protein levels of p53 are elevated in the liver of patients with T2D and p53 is positively correlated with glucose levels after an OGTT and the HOMA index. These correlations support the preclinical results and indicate that high levels of p53 in the liver are associated with human insulin resistance. Taking into account that upregulated PCK1 in type 2 diabetic patients correlates with increased rates of gluconeogenesis in the liver of humans[53], it is plausible to assume that p53 is somehow involved in the PCK1-related metabolic disorders. Finally, O-GlcNAc has been also implicated in insulin resistance[54] and gluconeogenesis[38](see review[21]) and diabetic individuals display higher leukocyte O-GlcNAcylation than nondiabetic controls[55]. In agreement with those reports, we also find increased levels of OGT and GFAT in the liver of patients with T2D and a positive correlation of GFAT with the glycemia 2 h after an OGTT, indicating that O-GlcNAcylation of p53 participates in the deregulated glucose metabolism of diabetes.

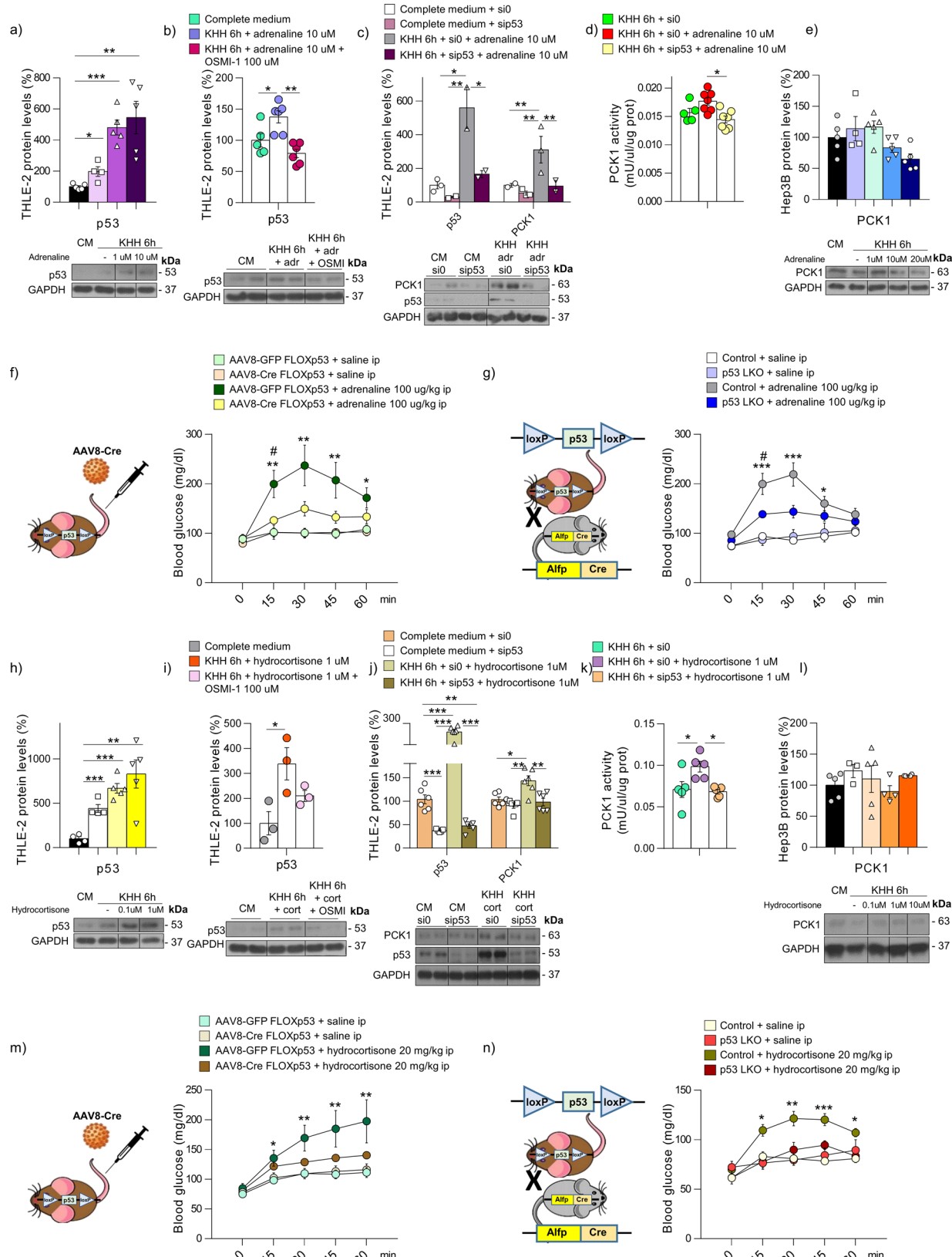

In summary, the present work unveils the role of p53 in the regulation of glucose homeostasis in physiological conditions, as well as a plausible pathophysiological role in the development of T2D in humans. Hepatic p53 binds to and regulates the transcriptional activity of PCK1 and this plays a pivotal role in the synthesis of glucose. Lack of p53 in hepatocytes impairs the gluconeogenic response during calorie restriction and the glucose excursion induced by glucagon, adrenaline, and adrenal glucocorticoids, while the upregulation of p53 in hepatocytes worsens insulin sensitivity. The stabilization of p53 upon fasting as well as its gluconeogenic action is dependent of its O-GlcNAcylation. Finally, protein p53 is elevated in the liver of type 2 diabetic

**Fig. 6 Hepatic p53 mediates the gluconeogenic action of adrenaline and cortisol. a** p53 levels in THLE-2 cells in the presence or absence of adrenaline ($n = 5$). **b** p53 levels in THLE-2 cells kept in complete medium (CM) or KHH + adrenaline in the presence or absence of OSMI-1 ($n = 6$). **c** Protein levels of p53 and PCK1 and **d** PCK1 activity in THLE-2 cells transfected with empty-siRNA or siRNA-p53 and then treated with KHH alone ($n = 5$) or KHH + adrenaline ($n = 7$). **e** PCK1 levels in Hep3B cells maintained in CM or KHH in the presence or absence of adrenaline ($n = 5$). **f** Blood glucose levels in p53 floxed mice injected with AAV8 expressing either GFP or Cre and then treated with saline or adrenaline ($n = 6$); * depicts differences compared to AAV8-GFP + saline, # depicts differences between AAV8-GFP and AAV8-Cre mice treated with adrenaline. **g** Blood glucose levels in p53 LKO mice and their control littermates treated with saline or adrenaline ($n = 6$). * depicts differences compared to control + saline, and # depicts differences between control and LKO mice treated with adrenaline. **h** p53 levels in THLE-2 cells incubated with different cortisol concentrations ($n = 5$). **i** p53 levels in THLE-2 cells kept in CM or KHH + hydrocortisone in the presence or absence of OSMI-1 ($n = 3$). **j** Protein levels ($n = 6$) of PCK1 and p53 and **k** PCK1 activity ($n = 5$) in THLE-2 cells transfected with empty-siRNA or siRNA-p53, and then treated with hydrocortisone. **l** PCK1 levels in starved Hep3B cells with different cortisol concentrations ($n = 5$). **m** Blood glucose levels in p53 floxed mice injected with AAV8 expressing either GFP or Cre and then treated with saline or hydrocortisone ($n = 7$ and 10). **n** Blood glucose levels in p53 LKO mice and their control littermates treated with saline or hydrocortisone ($n = 7$ and 12). Expression of GAPDH served as a loading control, and control values were normalized to 100%. Dividing lines indicate splicings within the same gel. Data were presented as mean ± standard error mean (SEM). * or # denotes $P < 0.05$, ** denotes $P < 0.01$, and *** denotes $P < 0.001$, determined in all cases by one-way ANOVA followed by Bonferroni post hoc testing. "$n$" denotes independent animals or cell culture wells. Source data are provided as a Source Data file.

patients and positively correlates with glucose and HOMA index. Altogether, our data uncover the physiological role of p53 as a hub integrating the signaling pathways influencing insulin sensitivity and liver responses to counterregulatory hormones in terms of glucose homeostasis.

## Methods

**Animals**. Animal experiments presented in this manuscript were approved by the Faculty Animal Committee at the University of Santiago de Compostela, and the experiments were performed in agreement with the Rules of Laboratory Animal Care and International Law on Animal Experimentation. Animals were fed with a standard diet (Scientific Animal Food & Engineering, proteins 16%, carbohydrates 60%, and fat 3%) and tap water ad libitum unless otherwise indicated. The number of animals used in each experiment is indicated in the corresponding figure legend. Eight weeks old male WT mice (C57BL/6), mice that carrying floxed p53 alleles (with a C57BL/6 background), and conditional p53 liver KO mice were housed in air-conditioned rooms (22–24 °C) under a 12:12 h light/dark cycle and controlled conditions of humidity (40%). p53 floxed mice were obtained from The Jackson Laboratory. Mice lacking p53 in the liver were generated in our laboratory crossing p53 floxed mice with Alfp-Cre mice, which express the Cre-recombinase open reading frame (ORF) under the control of both the mouse albumin regulatory elements and the α-fetoprotein enhancers (Alfp-Cre transgene), a configuration that mimics the genomic organization of the mouse albumin gene[56]. Animals were sacrificed and tissues were removed rapidly and immediately frozen in dry ice. Tissues were kept at −80 °C until their analysis.

### Animal experiments

*Effect of nutritional status on hepatic p53.* WT animals were fed either (1) ad libitum, (2) fasted for 6, 12, and 24 h, or (3) refed ad libitum for 24 or 2 h after 24 h starvation.

*Effect of leptin on hepatic p53 protein levels in vivo.* WT animals were treated with saline or recombinant leptin (Sigma-Aldrich, L-4146) at a dose of 0.5 mg kg⁻¹ of body weight every 12 h for 3 days by intraperitoneal injection (IP). WT mice were divided into three groups: (a) IP saline fed ad libitum chow diet, (b) IP saline while fasting for a period of 24 h, and (c) IP leptin while fasting for a period of 24 h.

*Effect of glucose on hepatic p53 protein levels in vivo.* WT mice were divided into three groups: (a) fed ad libitum chow diet, (b) fasting for a period of 24 h, and (c) fed with sucrose ad libitum during 24 h to maintain normal blood glucose levels and liver weight in the absence of other nutrients as previously described[18]. Blood glucose and body weight were measured at the beginning and after 24 h. To control whether the osmotic pressure could affect p53 levels, mannitol (D-mannitol; Merck, M4125) (which has a potent osmotic diuretic action) was administered to mice subjected to a fasting period of 24 h. Mice were fed with mannitol cubes ad libitum during 24 h.

*Glucagon administration.* WT mice, hepatic p53 deficient mice, and glucagon receptor KO mice (Gcgr −/− mice) were injected IP with 200 μg kg⁻¹ body weight of glucagon diluted in saline (Sigma-Aldrich, #G1774) and after 30 min, blood glucose levels were measured[57].

*Adrenaline administration.* WT mice and hepatic p53 deficient mice, were injected IP with 100 μg kg⁻¹ body weight of L-adrenaline diluted in saline (ThermoFisher, L04911).

*Cortisol administration.* WT mice and hepatic p53 deficient mice, were treated IP with 20 mg kg⁻¹ body weight hydrocortisone 21-hemisuccinate sodium salt diluted in saline (Santa Cruz Biotechnology, sc-250130).

*Hepatic insulin signaling in vivo.* To study the insulin signaling specifically in the liver, mice were fasted for 6 h and then anesthetized by an IP of ketamine (100 mg kg⁻¹ body weight)/xylazine (15 mg kg⁻¹ body weight). Adequacy of the anesthesia was ensured by loss of pedal reflexes. The abdominal cavity of the mice was opened, and 125 μl of samples containing 5 units of insulin (Actrapid, Novo Nordisk) diluted in saline were injected into the inferior cava vein. Sham injections were performed with 125 ml of saline. Samples of liver were harvested 2 min after injection[58].

*Calorie restriction.* Five days before initiation of calorie restriction, 8-week-old mice were placed in individual cages and fed the chow diet ad libitum. During these days of acclimation, food intake was monitored to determine the average amount of food consumed daily by each mouse. Thereafter, mice were randomly separated into two groups: one group continued to receive the chow diet ad libitum while the other group was subjected to 60% calorie restriction. Each mouse subjected to calorie restriction was fed at 6 p.m. every day with an amount of food equal to 40% of their daily food intake during the week of acclimation. Body weight and blood glucose were measured daily at 5:30 p.m. before feeding[59]. On the third day of calorie restriction, blood glucose levels were measured at 8 p.m. to assess the capacity of control mice and mice without hepatic p53 to recover normal blood glucose levels after feeding their daily food ration. Finally, mice were sacrificed at 5:30 p.m. (before feeding) on the fourth day of calorie restriction to collect blood and liver for analyses. NEFAs (WAKO, #434-91795) and ketone bodies (WAKO, #413-73601 and #415-73301) were assessed using commercial kits based on the colorimetric method.

*Postprandial glucose tolerance test.* Mice were starved overnight and then refed ad libitum for 240 min. Blood glucose, food intake, and body weight were measured at 0, 30, 60, 120, and 240 min after allowing them to eat ad libitum. To generate insulin resistance without changes in body weight and fatty liver phenotype[60], mice were fed with very HFD (60% fat, Research Diets, D12492) for 4 days. Then, mice were fasted overnight and refed ad libitum with very HFD, and blood glucose, food intake, and body weight were measured at indicated times.

*Glucose, insulin, pyruvate, and glycerol tolerance tests.* Basal blood glucose levels were measured after an overnight fast (12 h) for the GTT, PTT, and GlyTT; and after 6 h for the ITT, with a Glucocard Glucometer (ARKRAY, USA). GTT, ITT, PTT, and GlyTT were done after an IP of either 2 g kg⁻¹ D-glucose (Sigma-Aldrich, G8270), 0.35 U kg⁻¹ insulin (Actrapid, Novo Nordisk), 1.25 g kg⁻¹ sodium pyruvate (Sigma-Aldrich, P2256), and 1 g kg⁻¹ glycerol (Sigma-Aldrich, G5516), and area under the curve values were determined.

*In vivo virogenetic procedures.* To achieve a specific effect on the liver, viruses were injected into the tail vein. As such, mice were held in a specific restrainer for intravenous injections: Tailveiner (TV-150, Bioseb, France). Injections into the tail vein were carried out using a 27 G × 3/8″ (0.40 mm × 10 mm) syringe. Mice were injected with 100 μl of viral vectors diluted in saline. For the downregulation of p53 specifically in the liver, associated adenoviruses serotype 8 (AAV8) were injected in p53 floxed mice. AAV8-GFP and AAV8-Cre ($1 \times 10^{10}$VG ml⁻¹) (AAV8-GFP #SL100,833; AAV8-Cre #SL100,835 SignaGen Laboratories, USA) were injected into 4 weeks old p53 floxed mice, and 1 month after virus injection, experiments were performed. To rescue the expression of p53 specifically in the liver of the p53 liver-knockdown mice (floxed p53 mice previously treated with

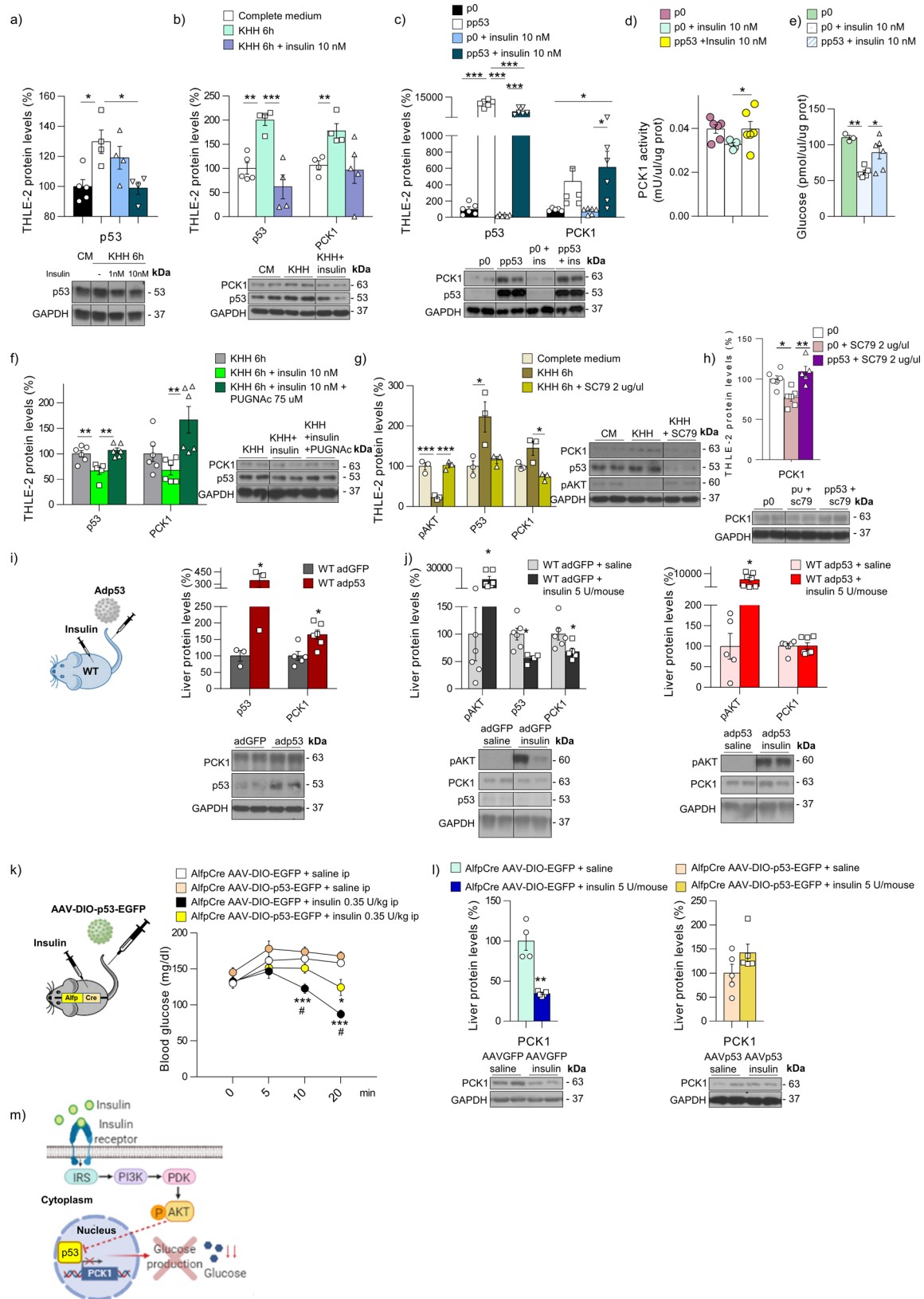

AAV8-Cre virus), these mice were injected with an adenoviral vector expressing p53 (SignaGen Laboratories, USA, ref # SL100,777) and controls were injected with an adenovirus encoding GFP (SignaGen Laboratories, USA, ref # SL100,833) $(1 \times 10^8 \text{ VG ml}^{-1})$[61]. Viruses were injected 1 day before the beginning of the calorie restriction experiment. To recover the hepatic expression of p53 in our conditional p53 liver KO mice (Alfp-Cre p53 floxed mice), animals were injected into the tail vein with Cre-dependent associated adenoviruses AAV8-DIO-p53-

EGFP and controls with AAV8-DIO-EGFP $(1 \times 10^8 \text{ VG ml}^{-1})$ (Vector Builder). Viruses were injected 1 month before the beginning of the experiments. To overexpress p53 specifically in hepatocytes, Alfp-Cre mice were injected into the tail vein with Cre-dependent associated adenoviruses AAV-DIO-P53-EGFP, and controls with AAV-DIO-EGFP $(1 \times 10^8 \text{ VG ml}^{-1})$. Viruses were injected 1 month before performing the experiments. Overexpression of hepatic p53 in WT mice to evaluate its effect on insulin signaling was achieved by tail vein injection of the

**Fig. 7 Overexpression of hepatic p53 worsens insulin sensitivity. a** p53 levels in THLE-2 cells maintained in complete medium (CM) ($n = 5$), KHH ($n = 4$), or KHH + insulin ($n = 4$). **b** Protein levels of p53 and PCK1 in THLE-2 cells maintained in CM, KHH, or KHH + insulin ($n = 5$). **c** Protein levels of p53 and PCK1 in THLE-2 cells transfected with empty plasmid or plasmid encoding p53 and then treated with insulin ($n = 6$). **d** PCK1 activity ($n = 6$) and **e** glucose production (p0 $n = 4$; p0 + insulin 10 nM $n = 5$; and pp53 + insulin 10 nM $n = 6$) in THLE-2 cells transfected with empty plasmid or plasmid encoding p53 cells in the absence or presence of insulin. **f** Protein levels of p53 and PCK1 in THLE-2 cells are maintained in KHH in the presence or absence of insulin and PUGNAc ($n = 6$). **g** Protein levels of pAKT, p53, and PCK1 in THLE-2 cells maintained in CM ($n = 3$) or KHH in the absence ($n = 3$) or presence of SC79 ($n = 4$). **h** PCK1 levels in THLE-2 cells transfected with empty plasmid or plasmid encoding p53 and then treated with SC79 ($n = 6$). **i** Protein levels of p53 and PCK1 in mice injected with empty adenovirus or p53 adenovirus (Adp53) ($n = 3$ and 6 per group). **j** Protein levels of pAKT, p53, and PCK1 in control mice or mice over-expressing p53 and then treated with saline or insulin into the cava vein ($n = 6$ and 7). **k** Blood glucose levels after IP insulin treatment ($n = 8$ and 10) and **l** protein levels of PCK1 in Alfp-Cre± mice injected with AAV8-DIO expressing either GFP or p53 and then treated with saline or insulin into the cava vein ($n = 4$ and 5).* depicts differences between AAV-EGFP + saline and AAV-EGFO + insulin and # depicts differences between AAV-EGFP and AAV-p53 treated with adrenaline. **m** Schematic representation of the role of p53 in insulin-induced gluconeogenesis suppression. Expression of GAPDH served as a loading control, and control values were normalized to 100%. Dividing lines indicate splicings within the same gel. Data were presented as mean ± standard error mean (SEM). * and # denotes $P < 0.05$, ** denotes $P < 0.01$, and *** denotes $P < 0.001$, determined by two-tailed Student's $t$-test (**i**, **j**, and **l**) or one-way ANOVA followed by Bonferroni post hoc testing (**a**, **b**, **c**, **d**, **e**, **f**, **g**, **h**, and **k**). "$n$" denotes independent animals or cell culture wells. Source data are provided as a Source Data file.

---

adenoviral vector encoding p53 (SignaGen Laboratories, USA, ref # SL100,777). WT mice injected with an adenovirus expressing GFP were used as control. Viruses were injected 4 days before the experiment. To downregulate OGT in the liver of WT mice, animals were injected into the tail vein with pLKO.puro.IRES.GFP lentiviral particles containing an shRNA against *Ogt* or *Luciferase* as a control. For the generation of lentiviral particles, $8 \times 10^6$ HEK293T cells were transfected using PEI, (Polyethylenimine; Sigma-Aldrich, ref#408727) and 20 μg of pLKO.shRNA plasmids alongside 10 μg of psPAX2 and pMD2.G packaging mix. Forty-eight and seventy-two hours later, supernatants were collected, and lentiviral particles were concentrated using 0.22 μm pore size centrifugal filter units (Amicon, ref#UFC903024). The following shRNA target sequences were used:

shOgt#1: 5′-CCCATTTCTTTCAGCAGAAAT-3′
shOgt#2: 5′-GCTCTTAATATGCCCGTTATT-3′
shLuciferase: 5′-CCTAAGGTTAAGTCGCCCTCG-3′

*Determination of HGP.* HGP measurement was performed in conscious unrestrained catheterized mice. Catheters (BTSIL-025, Instech Laboratories Inc, Plymouth Meeting, PA) were surgically implanted under isoflurane anesthesia, 7 days prior to the experiment in the right jugular vein and exteriorized above the neck using a vascular access button (VAB62BS/25, Instech Laboratories Inc.). Catheters were washed every day using heparinized saline (NaCl 0.9%, 200 IU/ml heparin). Mice receiving the last part of the food restriction at 7 p.m. the day before, no fasting was needed the day of perfusion. Mice then received a 90-min infusion of [3-³H] glucose (0.05 μCi/min) (Perkin Elmer, Walthman, MA). Blood was sampled from the tail vein at 90, 100, and 110 min (10 μl) and treated with 0.3 N ZnSO₄ and 0.3 N Ba(OH)₂. Glucose concentration was measured using the glucose oxidase method (GLU, Roche Diagnostics, Rotkreuz, Switzerland) and measurements of [3-³H] glucose was done with or without overnight drying for the evaluation of tritiated water for glycolysis.

*Hematoxylin/eosin staining and immunohistochemistry.* Liver samples were fixed 10% formalin buffer for 24 h, and then dehydrated and embedded in paraffin by a standard procedure. Sections of 3 μm were prepared with a microtome and stained using a standard Hematoxylin/Eosin Alcoholic (BioOptica) procedure according to the manufacturer's instructions. Alternative sections of paraffin were used for immunohistochemistry detection of CC3 and ki67. Up to ten animals per experimental group were used and three pictures per image section were taken.

*Chromatin immunoprecipitation in liver samples.* About 5 mg of liquid N₂-frozen tissue was finely minced on ice and disaggregated in a Dounce homogenizer [KIMBLE Dounce tissue grinder set, SIGMA #D8938, pestle A (10 times) and pestle B (20 times)] using 1.5 ml of PBS-P [1X PBS supplemented with proteinase inhibitors (Roche, Cat No. 000000011873580001–Merck)]. About 1 ml of cell suspension was crosslinked by adding 100 μl of freshly prepared 11% formaldehyde solution (50 mM Hepes-KOH, pH 7.5, 100 mM NaCl, 1 mM EDTA, 0.5 mM EGTA, and 11% formaldehyde) and rocked at room temperature for 8 min. Cross-linking was stopped by adding glycine to a final concentration of 0.125 M for 5 min, and the suspension was centrifuged at 9000x*g* for 5 min at 4 °C, washed with PBS-P1X, centrifuged, and the pellet flash-frozen in liquid N₂. The tissue pellet was resuspended twice in 2 ml of cell lysis buffer [20 mM Tris-HCl pH 8.0, 85 mM KCl, 0.5% Igepal (Sigma, Cat No. I8896) supplemented with 1X proteinase inhibitors], vortexed, and left on ice for 10 min, followed by centrifugation at 9000x*g* for 5 min at 4 °C. Finally, the pellet was disaggregated using a Dounce homogenizer (pestle B, 20 times) in 200 μl of nuclear lysis buffer (10 mM Tris-HCl, pH 8.0, 1 mM EDTA, and 1% SDS, supplemented with 1X Protease inhibitor), left on ice for 10 min, vortexed, and flash-frozen in liquid N₂.

About 200 μl of nuclear extracts were diluted with 800 μl of TE, and 500 μl of the mix were sonicated in 1.5 ml DNA LoBind Tube 1.5 ml (Eppendorf, Cat. No.022431021) using the Bioruptor™ Plus sonication device (Diagenode) with high power mode for 30 cycles (sonication cycle: 30 sec ON, 30 sec OFF) followed by centrifugation at 10,000x*g* for 10 min at 4 °C. The supernatant containing sonicated chromatin was aliquoted and kept at –80 °C until further use. To check cross-linking efficiency, 25 μl of the sonicated chromatin was reverse cross-linked at 65 °C overnight in 125 μl of elution buffer (50 mM Tris-HCl, pH 8.0, 10 mM EDTA, and 1% SDS) and NaCl (0.2 M). About 250 μl of TE and RNase A (0.2 mg/ml) were added and incubated at 37 °C for 2 h. Proteinase K (0.2 μg/ml) was added and incubated at 56 °C for 2 h. Eluted DNA was purified using Quiagen QIAquick PCR purification kit (Cat. No. 28104), and chromatin fragmentation of DNA (100–500 ng) was checked in a 1% agarose gel.

ChIP was then carried out using the Magna ChIP A/G Chromatin Immunoprecipitation Kit (Merck, Cat no. 17-10085) following the manufacturer´s instructions. In brief, the samples were immunoprecipitated with 5 μl of anti-P53 (1C12) Mouse mAb (2524, Cell Signaling Technology) overnight at 4 °C, washed once with low-salt, high-salt, LiCl buffer, and TE buffer, and the eluted DNA was reverse-crosslinked and purified using PCR purification kit (Qiagen). The recovered DNA was subjected to PCR amplification and the abundance of target genome DNA was calculated as the percentage of input. Primers used for the PCK1 promoter region are the following: Forward (CAACAGGCAGGGTCAAAGT) and Reverse (GCACGGTTTGGAACTGACTT) (Supplementary Table 9).

*Cell cultures and transfections.* THLE-2 cells, a human liver cell line, was purchased from ATCC (The Global Bioresource Center; CRL-2706). Cells were cultured in Bronchial Epithelial cell Basal Medium (BEBM) supplemented with a growth factor BulleKit (Lonza/Clonetics Corporation, USA, BEGM Bullet Kit; CC-3170), 70 ng ml⁻¹ phosphoethanolamine (Sigma; #P05039), 5 ng ml⁻¹ epidermal growth factor, 10% fetal bovine serum (FBS) (Gibco; #1027016), and 1% Glutamine-Penicillin-Streptomycin solution (Sigma-Aldrich; G6784). THLE-2 cells were grown on culture plates pre-coated with a mixture of 0.01 mg/ml fibronectin (Gibco, #33010018), 0.01 mg/ml bovine serum albumin (BSA; Sigma, #A4503), and 0.03 mg ml⁻¹ collagen type I (Santa Cruz, #sc-136157). A total of, $1.5 \times 10^5$ cells were seeded in a six-well plate for all experiments. Hep3B cells, a KOp53 cell line derived from a human p53 deficient liver, were obtained from ECACC (European Collection of Authenticated Cell Cultures, Sigma; #86062703) and maintained as a monolayer culture in Minimum Essential Medium Eagle (EMEM) (Sigma-Aldrich, #M2279) supplemented with 10% FBS, 1% Glutamine-Penicillin-Streptomycin solution (Sigma, #G6784), and 1% nonessential amino acid (NEAA) (Sigma-Aldrich, #M7145) (growth medium). A total of, $3 \times 10^5$ cells were seeded in a six-well plate for all experiments. Both cell lines were maintained at 37 °C in a humidified atmosphere containing 5% CO₂ and were routinely tested for mycoplasma.

To knockdown the expression of p53 in THLE-2 cells, cells were transfected with specific small-interference RNA (siRNA) to p53 (Dharmacon, #L-003329-00). Non-targeting siRNA was used as negative control (Dharmacon, #D-001810-10-05). The transfection was performed using Dharmafect 1 reagent (Dharmacon, #t-2001-03) following the protocol: 50 pmol of the sip53 diluted in 200 μl of Opti-MEM (Life Technologies #31985-070) was mixed with 6.5 μl of Dharmafect 1 diluted in 193.5 μl of Opti-MEM; the mixture was added into each well, resulting in a final volume of 1.5 ml with BEGM complete medium for 6 h. After that, the medium was replaced with fresh BEGM until indicated treatments were performed. Forty-eight hours after the plasmidic p53 transfection, cells were collected for protein extraction.

To upregulate p53 protein levels in THLE-2 cells, cells were transfected with pCMV-Neo-Bam and pCMV-Neo-Bam p53 wt (Addgene plasmid #16440 and

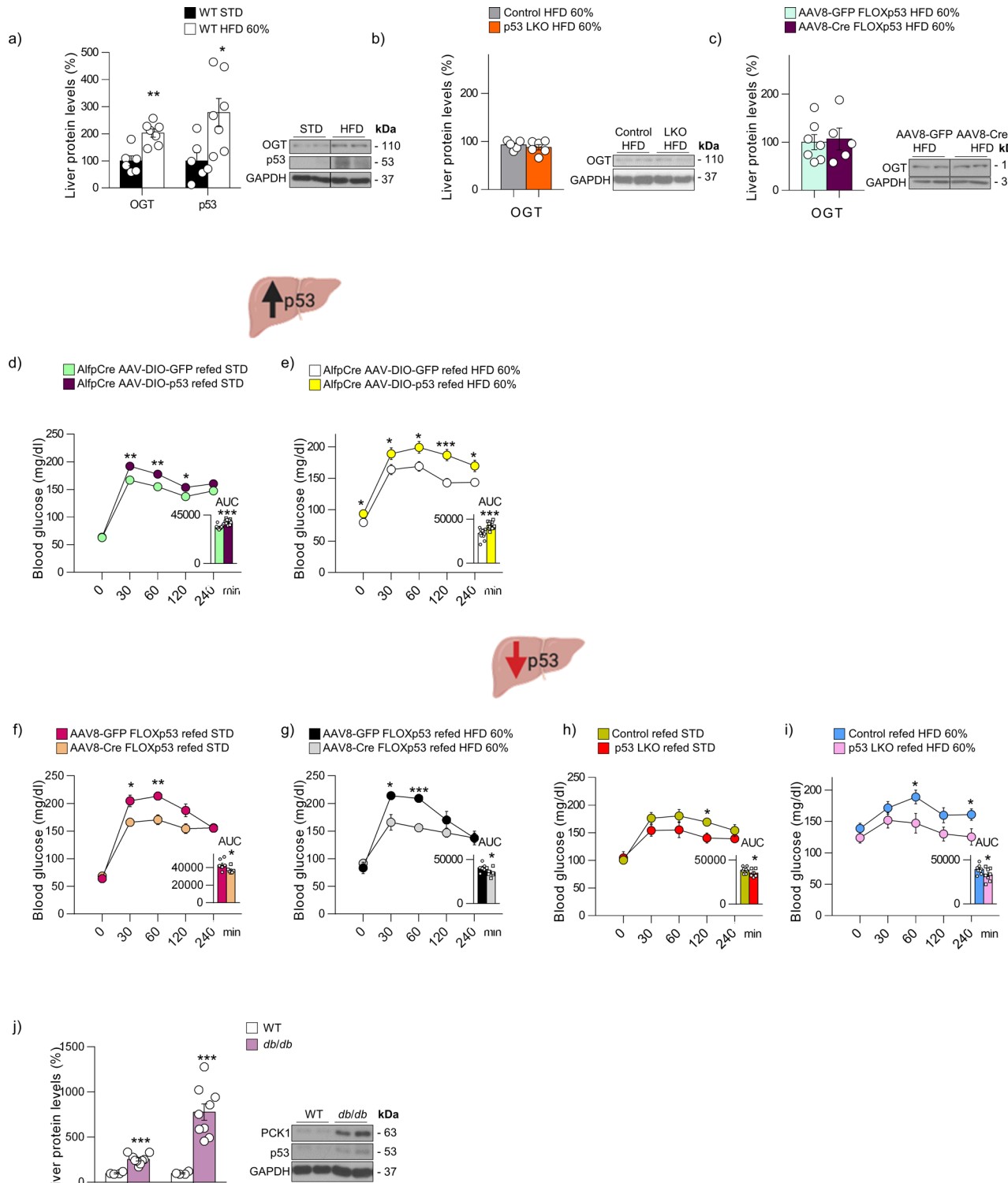

**Fig. 8 Hepatic p53 affects postprandial hyperglycemia and is elevated in models of insulin resistance. a** Hepatic OGT and p53 protein levels in WT mice fed with standard diet or high-fat diet (HFD) 60% for 4 days ($n = 7$). Hepatic OGT protein levels in (**b**) control mice and p53 LKO mice fed with HFD 60% for 4 days ($n = 6$) and **c** FLOXp53 mice injected with AAV8 expressing either GFP ($n = 7$) or Cre ($n = 5$) and fed with HFD 60% for 4 days. Blood glucose levels in Alfp-Cre± mice injected with AAV-DIO expressing either GFP or p53 refed with **d** chow diet ($n = 11$ and 12) and **e** HFD after overnight fasting ($n = 11$ and 12). Blood glucose levels in FLOXp53 mice injected with AAV8 expressing either GFP or Cre refed with **f** chow ($n = 6$ and 8) diet and **g** HFD after overnight fasting ($n = 5$ and 6). Blood glucose levels in control and p53 LKO ($n = 6$) mice refed with **h** chow ($n = 5$ and 7) diet and **i** HFD ($n = 7$) after overnight fasting. **j** Hepatic p53 and PCK1 protein levels in WT ($n = 4$) and db/db mice ($n = 9$). The area under the curve (AUC) is provided. Expression of GAPDH served as a loading control and control values were normalized to 100%. Dividing lines indicate splicings within the same gel. Data were presented as mean ± standard error mean (SEM). * denotes $P < 0.05$, ** denotes $P < 0.01$, and *** denotes $P < 0.001$, determined by two-tailed Student's t-test. "n" denotes independent animals. Source data are provided as a Source Data file.

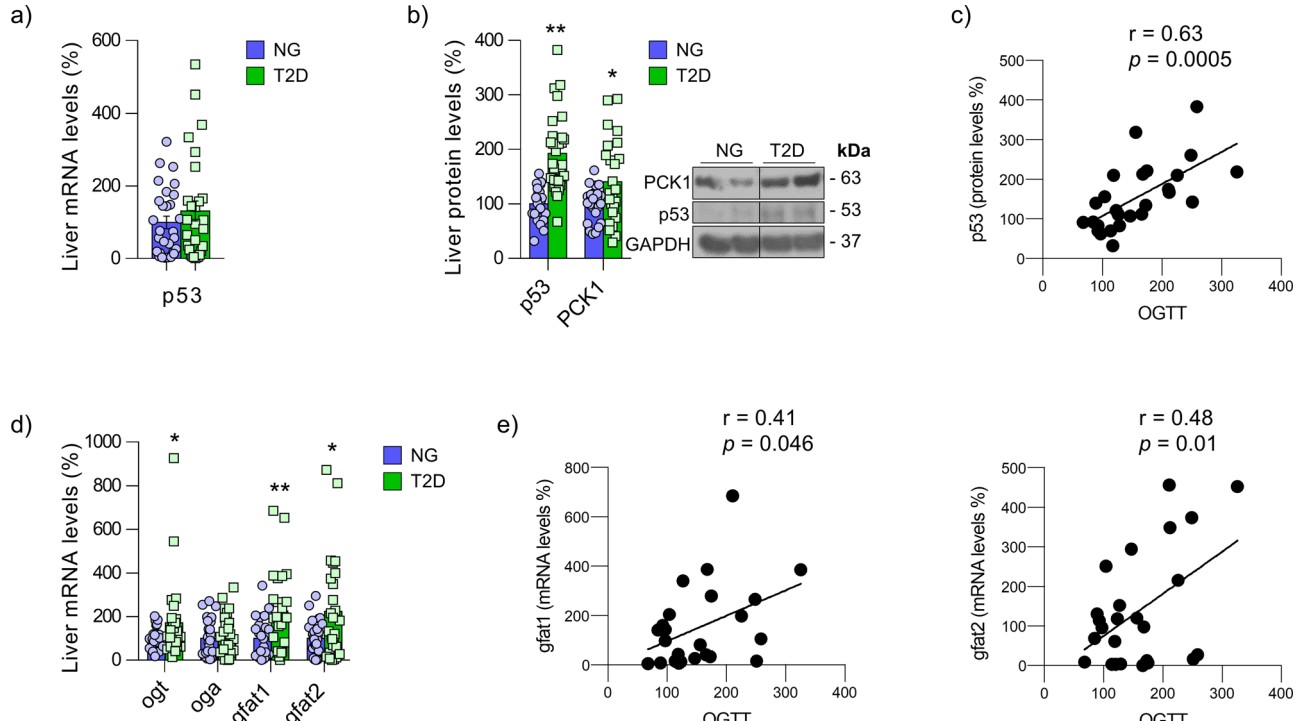

**Fig. 9 p53 protein levels are increased in the liver of patients with T2D.** Obese patients were further subclassified according to their normoglycemia (NG) or type 2 diabetes (T2D) ($n = 30$ human patients per group). **a** Hepatic *p53* mRNA levels in NG or T2D patients. **b** Hepatic p53 and PCK1 protein levels in patients with NG and T2D. **c** Correlation between p53 protein levels and OGTT. **d** Expression of *OGT*, *OGA*, *GFAT1*, and *GFAT2* in patients with NG and T2D. **e** Correlation between *GFAT1* and *GFAT2* expression and OGTT. Expression of GAPDH (western blot) and HPRT (qRT-PCR) served as a loading control, and control values were normalized to 100%. Dividing lines indicate splicings within the same gel. Data were presented as mean ± standard error mean (SEM). * denotes $P < 0.05$, ** denotes $P < 0.01$, and *** denotes $P < 0.05$ determined by two-tailed Student's *t*-test. Source data are provided as a Source Data file.

Addgene plasmid # 16434). To upregulate mutated p53 in Ser149 (p53-S149A), which can not be *O*-GlcNAcylated, cells were transfected with His-p53-S149A plasmid, kindly provided by Dr. Jin Won Cho, Yonsei University, Korea. To upregulate PCK1, cells were transfected with pCMV6-PCK1-human-tagged-ORF-clone (Origene plasmid #RC204758). Lipofectamine 2000 (Invitrogen, #11668-019) was used to transfect cells with the following protocol: 4 μl of Lipofectamine 2000 diluted on 150 μl of Opti-MEM mixed with 2.5 μg of DNA diluted on 150 μl of Opti-MEM. This mixture was incubated in a final volume of 1.5 ml of BEGM complete medium for 6 h. After that, the medium was replaced with a fresh complete BEGM medium until cells were subjected to the indicated treatments. p53 was upregulated during 24 h before cells were collected for protein or mRNA extraction. To rescue the p53 expression in Hep3B cells, cells were transfected with a DNA plasmid encoding p53. Lipofectamine 2000 was used to transfect the cells with the following protocol: 8 μl of Lipofectamine 2000 diluted on 150 μl of opti-MEM mixed with 8 μg of DNA diluted on 150 μl of opti-MEM. This mixture was incubated in a final volume of 1,5 ml of Minimum Essential Medium Eagle complete medium supplemented with FBS 10% for 6 h. After that, the medium was replaced with a fresh complete EMEM medium until cells were subjected to the indicated treatments. p53 was upregulated during 24 h before cells were collected for protein or RNA extraction.

*Cell treatments.* Cells were starved for 6 h in Krebs–Henseleit–HEPES buffer (KHH; composition: 120 mmol l$^{-1}$ NaCl, 4.7 mmol/l KCl, 2.5 mmol/l CaCl$_2$, 1.2 mmol l$^{-1}$ MgSO$_4$, 1.2 mmol l$^{-1}$ KH2PO$_4$, 25 mmol l$^{-1}$ NaHCO$_3$, and 25 mmol/l HEPES pH 7.4.), a fasting medium with neither nutrients nor hormones[62]. To study the regulation of p53 protein expression by glucose in vitro, cell medium from untreated cells were changed to KHH, in the presence or absence of glucose (D-glucose, Sigma-Aldrich, #G8270) 1 or 10 mM. For the following experiments, the dose of 10 mM glucose was used. Cells were also treated with mannitol 10 mM to study the effect of osmotic pressure on p53 protein levels. To study the role of *O*-GlcNAcylation of p53, we treated cells with PUGNAc (Sigma-Aldrich, #A7229) (an inhibitor of OGA, the enzyme which removes *O*-GlcNAc from *O*-linked glycosylated proteins) 5 uM, 25 μM, and 75 μM. For the following experiments, a dose of 75 μM was used. Cells were also treated with OSMI-1 (Sigma-Aldrich, #SML1621) (an inhibitor of OGT, which catalyzes the addition of a single *O*-GlcNAc through an O glycosidic linkage to a serine or threonine and an S glycosidic linkage to cysteine residues of intracellular substrate proteins) 10, 40, and

100 μM. For the following experiments, the dose of 100 μM was used. THLE-2 cells were also treated with vehicle or doxorubicin 25 nM (Tedec-Feji Farma) for 48 h [28].

To study the regulation by glucagon in vitro, cells were starved and treated with glucagon (Sigma-Aldrich, #G1774) 0.1, 10, and 25 nM, For the following experiments, the dose of 10 nM was used. Cells were also treated with forskolin (Santa Cruz Biotechnology; sc-3562) 2 μM in complete medium for 30 min. Forskolin is a cAMP activator that mimics the effect of glucagon activating its molecular pathway. To study the regulation by adrenaline (Thermofisher; #L04911) in vitro, cells were starved and treated with adrenaline 1, 10, and 20 μM[63]. The dose of 10 μM was used for the following experiments. To study the effect of cortisol, cells were starved and treated with hydrocortisone solution (Sigma-Aldrich, #H6909) 0.1, 1, and 10 μM[64]. To study the regulation by insulin in vitro, cells were starved and treated with insulin 1 and 10 nM[65]. For the following experiments, a dose of 10 nM was used. Cells were also treated with SC79 Akt Activator II (Merk Millipore, #123871) 2 μg μl$^{-1}$[66] for 6 h.

*Glucose measurement in cell culture medium.* Cells were incubated in glucogenic medium (KHH medium supplemented with lactate 20 mM and pyruvate 2 mM) for indicated times in the absence or presence of forskolin 2 μM or insulin 10 nM. Glucose released to the media was quantified using the High Sensitivity Glucose Assay Kit (Sigma-Aldrich, #MAK181-1KT). Results were normalized by the amount of protein of cell lysate [67].

*Oxaloacetate assay kit.* Cells were incubated in complete medium or KHH for 6 h. Cellular oxaloacetate content was quantified using the Oxaloacetate Assay Kit (Sigma-Aldrich, #MAK070-1KT). Results were normalized by the amount of protein of cell lysate.

*Phosphoenolpyruvate carboxykinase activity assay.* PCK1 activity was measured in cell and tissue samples following manufacturer instructions (Biovision, #K359-100). Briefly, samples were homogenized with 200 μl ice-cold Assay Buffer and kept for 10 min on ice. After centrifuging them for 10 min at 10000x*g* at 4 °C, the supernatant is collected, and protein level concentration and PCK activity are measured separately. In this assay, PCK is coupled with a set of enzymes that convert PEP and carbonate into a series of intermediates and hydrogen peroxide, which in turn, reacts with a probe and converted generating a colorimetric signal

(OD: 570 nm). The color intensity is directly proportional to the amount of active PCK present in samples.

*Protein immunoprecipitation.* p53 immunoprecipitation was performed using a Pierce™ Direct IP Kit (ThermoFisher Scientific, 26148). This IP method can be used to cross-link the primary antibody to a resin. Immobilized primary antibody results in purified antigen free from antibody denatured chains which turns out crucial since heavy chain matches with p53 molecular weight in western blot analysis. According to the manufacture's recommendations, 5 μg of primary antibody against human p53 (DO-1) (Santa Cruz Biotechnology, sc-126) was chemically coupled to the AminoLink resin for 120 min. An additional tube was used to cross-link 5 μg of an appropriate isotypic control for primary antibody (normal mouse IgG$_{2a}$, Santa Cruz Biotechnology, sc-3878). About 750 μg of total protein THLE-2 samples, pretreated with lysis buffer supplemented with protease and phosphatase inhibitors and PUGNAc 1 μM to avoid the loss of posttranslational modifications[23] and precleared, were added to the antibody-coupled resin and gently rotated at 4 °C overnight to allow binding of the antigen with the immobilized antibody. The sample was then washed with the provided solutions and eluted with a final volume of 40 μL of Elution Buffer in a collection tube with a proper amount of 1 M Tris, pH 9.5 to neutralize the pH and to avoid antigen degradation. The final sample was completed with sample buffer, boiled to 95 °C for 5 min, and analyzed by western blot analysis as normal. The membrane was incubated first with the anti-O-linked *N*-acetylglucosamine primary antibody [RL2] (Abcam, ab2739) and then a β-mercaptoethanol based stripping procedure was performed to check for total immunoprecipitated p53 using the anti-p53 antibody from Cell Signaling (Supplementary Table 8).

*Luciferase assay and chromatin immunoprecipitation in cells.* For luciferase assays, Hep3B cells were transfected in six-well plates containing 6 μl of jetPRIME Poly-plus transfection reagent (PolyPlusTransfection, Illkirch, France), 2 μg of pRSV5 control or pRSV5-p53 overexpression vectors, 1 μg of each luciferase reporter plasmids pADTrack, pADTrack-1330-pck1, and pADTrack-490-pck1 (containing the −1330 and the −490 bp of the promoter region of the *PCK1* gene), and 50 ng of pRL-TK-Renilla (as transfection control) for 48 h. Cells were lysed in buffer (200 μl lysis buffer, Promega Corporation, WI, USA) and luciferase was measured in a Mithras LB 940 apparatus (Berthold Technologies, Bad Wildbad, Germany).

ChIP assays were performed in AML12 cells using the Upstate protocol as described previously[68]. Diluted soluble chromatin fractions were immunoprecipitated with 1 μg polyclonal anti-p53 antibody (Cell Signaling, MA, USA), or control human immunoglobulin G (IgG) (Sigma-Aldrich, St. Louis, USA). The histone-DNA crosslinks were reversed by a 4-h incubation at 65 °C. PCR was used to analyze the DNA fragments from ChIP assays. The PCR was run for 1 min at 95, 58, and 72 °C within each cycle, for 35 cycles in total. The three pairs of PCK1 primers were: the proximal promoter pair (1) (forward, −477/−457 bp) 5′-TGGCTCAGAGCTGAATTTCC-3′, (reverse, −312/−292 bp) 5′-GCAGGCTCTTGCCTTAATTG-3′; PCR product 185 bp). (2) (forward, −280/−260 bp) 5′-CAACAGGCAGGGTCAAAGTT-3′, (reverse, −145/−125 bp) 5′-GCACGGTTTGGAACTGACTT-3′; PCR product 155 bp) (3) (forward, −122/−107 bp) 5′-CCATGGCTATGATCCAAAGG-3′, (reverse, −1/+18 bp) 5′-CAGAGGGAAGGCCAACTGT-3′; PCR product 140 bp). PCR products were quantified using the ImageJ software and represented as mean ± SEM.

Primers sequences are detailed in Supplementary Table 9.

*Pyruvate-1-$^{13}$C pulse-chase analysis by NMR.* Three different conditions were tested in hepatocytes: control cells, p53-WT over-expressing cell line, and p53-S149A mutant over-expressing hepatocytes. For each condition, 20–30 million cells were grown. To study the role of p53 gluconeogenic metabolic flux in vitro, cell medium from untreated cells were changed to KHH, in the presence of pyruvate-1-$^{13}$C (110 mg/L of media) (Merck, 490709). Cells were incubated for 24 h, washed in PBS twice, and harvested in the same buffer. Cell metabolic extraction was performed as explained[69]. The hydrophilic metabolites were resuspended in 300 μl of D$_2$O with 0.11 mM of Sodium trimethylsilylpropanesulfonate (DSS, internal reference). All NMR experiments were recorded at 298 K on a Bruker 600 MHz (12 T) Avance III spectrometer equipped with a BBO probehead. For each sample, three different experiments were collected: (I) 1D $^1$H p3919gp with water signals suppression using a binomial 3-9-19 pulse with echo gradient pair, (II) 1D $^{13}$C zgpg30 with power-gated decoupling, and (III) phase-sensitive gradient-enhanced 2D 1H-13C HSQC (hsqcetgp). Chemical shift assignment was performed using standard compounds. Peak integration and quantification were done using Top-Spin 4.0.7 (Bruker Biospin GmbH) and in-house MatLab scripts.

*Human patients.* Liver samples were obtained from patients with severe obesity undergoing bariatric surgery (n = 60) at the Clínica Universidad de Navarra. Obesity was defined as a BMI ≥30 kg/m$^2$ and body fat percentage (BF) ≥35%. BMI was calculated as weight in kilograms divided by the square of height in meters, and BF was estimated by air-displacement plethysmography (Bod-Pod®, Life Measurements, Concord, CA, USA). Patients with obesity were subclassified into two groups [NG or T2D] following the criteria of the Expert Committee on the Diagnosis and Classification of Diabetes[70]. Inclusion criteria encompassed a complete diagnostic work-up including physical examination, laboratory

investigation, ultrasound echography, and liver biopsy consistent with the diagnosis of nonalcoholic fatty liver disease (NAFLD) according to the criteria of Kleiner and Brunt by an expert pathologist masked to all the results of the assays[71]. Features of steatosis, lobular inflammation, and hepatocyte ballooning were combined to obtain NAFLD activity score (NAS) (0–8)[71]. Exclusion criteria were: (a) excess alcohol consumption (≥20 g for women and ≥30 g for men); (b) the presence of hepatitis B virus surface antigen or hepatitis C virus antibodies in the absence of a history of vaccination; (c) use of drugs linked to NAFLD, including amiodarone, valproate, tamoxifen, methotrexate, corticosteroids, or antiretrovirals; (d) evidence of other specific liver diseases, such as autoimmune liver disease, hemochromatosis, Wilson's disease, or α-1-antitrypsin deficiency. Patients with T2D were not on insulin therapy or medication likely to influence endogenous insulin levels. It has to be stressed that patients with T2D did not have a long diabetes history (less than 2–3 years or even de novo diagnosis as evidenced from their anamnesis and biochemical determinations). All reported investigations were carried out in accordance with the principles of the Declaration of Helsinki, as revised in 2013, and approved by the Clínica Universidad de Navarra Ethical Committee responsible for research (protocol 2017.104). Written informed consent was obtained from all the participants.

*Quantitative reverse transcriptase PCR (qRT-PCR) analysis.* RNA was extracted using TRIzol reagent (ThermoFisher, USA) according to the manufacturer's instructions. Total RNA of 100 ng of total RNA was used for each RT reaction, and cDNA synthesis was performed using the SuperScript First-Strand Synthesis System (Invitrogen) and random primers. Negative control reactions, containing all reagents except the sample were used to ensure specificity of the PCR amplification. For analysis of gene expression, we performed real-time reverse-transcription polymerase chain reaction (RT-PCR) assays using SYBR green reagent (Agilent Technologies, USA). The PCR cycling conditions included an initial denaturation at 95 °C for 3 min followed by 40 cycles at 95 °C for 5 s and 60 °C for 32 s. The oligonucleotide-specific primers are shown in (Supplementary Table 7). For data analysis, the input value of gene expression was standardized to the HPRT value for the sample group and expressed as a comparison with the average value for the control group. All samples were run in duplicate, and the average values were calculated.

*Western blot analysis.* Blots were analyzed with ImageJ software 1.8.0. Total protein lysates from the liver (20 μg) and cells (2 μg) were subjected to SDS-PAGE, electrotransferred onto polyvinylidene difluoride membranes (Millipore), and probed with the indicated antibodies (Supplementary Table 8).

**Statistics and reproducibility.** Data were expressed as mean ± standard error mean (SEM). Statistical significance was determined by two-tail Student's *t*-test when two groups were compared or ANOVA and post hoc two-tailed Bonferroni test when more than two groups were compared. $P < 0.05$ was considered significant for all the analysis. The correlations between p53 and PCK1 protein and mRNA levels, and between p53, *gfat1*, and *gfat2* with OGTT were analysed by Pearson's correlation coefficients (*r*). Data analysis was performed using GraphPad Prism Software Version 8.0 (GraphPad, San Diego, CA) and Microsoft Excel. All the results shown in the manuscript are representative of, at least, two independent experiments with the same result.

**Reporting Summary.** Further information on research design is available in the Nature Research Reporting Summary linked to this article.

## Data availability
The authors declare that all data supporting the findings of this study are available within the article in the Supplementary information and Source Data file. Further information can be provided upon request. Source data are provided with this paper.

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

## Acknowledgements

This work has been supported by grants from FEDER/Ministerio de Ciencia, Innovación y Universidades-Agencia Estatal de Investigación (C.D.: BFU2017-87721; M.L.: RTI2018–101840-B-I00; R.N.: RTI2018-099413-B-I00; M.L.M.-C: SAF2017-87301-R; A.W.: RTI2018-097503-B-I00, and SAF2015-62588-ERC), Xunta de Galicia (M.L.: 2016-PG068; R.N.: 2015-CP080 and 2016-PG057), Fundación BBVA (R.N., A.W., G.S., and M.L.M.-C.), Programa Retos (M.L.M.-C: RTC2019-007125-1), Proyectos Investigación en Salud (M.L.M.-C: DTS20/00138), Fundación Atresmedia (M.L. and R.N.), Gilead Sciences International Research Scholars Program in Liver Disease (M.V.-R.), the Western Norway Regional Health Authority; Helse Vest RHF (J.F.), National Research Foundation of Korea Grant, Ministry of Science, ICT and Future Planning (J.W.C. and W.H.Y.: NRF-2016R1A5A1010764), Basque Department of Industry, Tourism and Trade (A.W.) and European Foundation for the Study of Diabetes (R.N. and G.S.), the German research foundation DFG (TRR152 and TRR296)/T.D.M.). The research leading to these results has also received funding from the European Community's H2020 Framework Program under the following grant: ERC Synergy Grant-2019-WATCH- 810331 to R.N. and ERC Consolidator Grant (865157-MYERIBO) to A.W. Centro de Investigación Biomédica en Red (CIBER) de Fisiopatología de la Obesidad y Nutrición (CIBERobn) and Centro de Investigación Biomédica en Red (CIBER) de Enfermedades Hepáticas y Digestivas (CIBERehd). CIBERobn and CIBERehd are initiatives of the Instituto de Salud Carlos III (ISCIII) of Spain which is supported by FEDER funds. We thank MINECO for the Severo Ochoa Excellence Accreditation to CIC bioGUNE (SEV-2016-0644).

## Author contributions

M.J.G-R., M.F.F., and U.F. carried out the experiments, analyzed data, developed analytical tools, and contributed to the discussion. A.R., M.V-R, C.V-D., S.S., G.B., F.L-O., D.F.-R., J.B., C.I., E.N., C.A., A.S., D.B., J.C., M.d.P.C.-V., M.G.-V., S.B.B., N.L., B.P., C.C., A.V., and S.T. carried out the experiments and analyzed data. T.D.M., J.F., D.G., M.F., G.S., S.H., W.H.Y., J.W.C., M.L.M.-C., R.P.-F., M.L., C.D., J.M. M., O.M., R.C., A.W., and G.F. contributed to discussion and designed experiments. M.J.G.-R and R.N. designed experiments and wrote the paper. R.N. serves as the guarantor.

## Competing interests

The authors declare the following competing interests: O.M. and G.B. are employees of ATLAS Molecular Pharma.

## Additional information

[1]CIMUS, University of Santiago de Compostela-Instituto de Investigación Sanitaria, Santiago de Compostela, Spain. [2]CIBER Fisiopatologia de la Obesidad y Nutrición (CIBERobn), Madrid, Spain. [3]Metabolic Research Laboratory, Clínica Universidad de Navarra and IdiSNA, Pamplona, Spain. [4]Liver Disease Laboratory, Center for Cooperative Research in Biosciences (CIC bioGUNE, Basque Research and Technology Alliance (BRTA), Bizkaia Technology Park, Derio, Spain. [5]CIBERehd, Instituto de Salud Carlos III, Madrid, Spain. [6]Department of Cell Physiology and Metabolism, Faculty of Medicine, University of Geneva, Geneva, Switzerland. [7]Diabetes Center of the Faculty of Medicine, University of Geneva, Geneva, Switzerland. [8]Precision Medicine and Metabolism Laboratory, CIC bioGUNE, Basque Research and Technology Alliance, Derio, Spain. [9]ATLAS Molecular Pharma S. L., Derio, Spain. [10]Proteomic Unit, Health Research Institute of Santiago de Compostela (IDIS), Santiago de Compostela, Coruña, Spain. [11]Institute for Diabetes and Obesity, Helmholtz Diabetes Center (HDC) at Helmholtz Zentrum München, German Research Center for Environmental Health (GmbH) and German Center for Diabetes Research (DZD), Oberschleissheim, Germany. [12]Department of Pharmacology, Experimental Therapy and Toxicology, Institute of Experimental and Clinical Pharmacology and Pharmacogenomics, Eberhard Karls University Hospitals and Clinics, Tübingen, Germany. [13]Hormone Laboratory, Haukeland University Hospital, Bergen, Norway. [14]Centro Nacional de Investigaciones Cardiovasculares (CNIC), Madrid, Spain. [15]Institute for Diabetes and Cancer (IDC) and Joint Heidelberg-IDC Translational Diabetes Program, Helmholtz Center Munich, Neuherberg, Germany. [16]Department of Systems Biology, Yonsei University, Seoul, Korea. [17]IKERBASQUE, Basque Foundation for Science, Bilbao, Spain. [18]CIMUS, University of Santigo de Compostela-Instituto de Investigación Sanitaria, Santiago de Compostela, Spain. [19]Nerve Disorder Laboratory, Center for Cooperative Research in Biosciences (CIC bioGUNE, Basque Research and Technology Alliance (BRTA), Bizkaia Technology Park, Derio, Spain. [20]Galician Agency of Innovation (GAIN), Xunta de Galicia, Santiago de Compostela, Spain. [21]These authors contributed equally: Marcos F. Fondevila, Uxia Fernandez. ✉email: ruben.nogueiras@usc.es

