## [Peer Review File · Nature Communications]

Reviewers' Comments:

Reviewer #1:

Remarks to the Author:

The paper examines the effect of hepatic p53 loss on gluconeogenesis, concluding that p53 is required to induce PCK1 expression to drive the gluconeogenic response. In general, the effects of liver specific p53 deletion on blood glucose levels in response to various stimuli is compelling and consistent. However, the proposed mechanism underpinning these observations are less convincing. Many of the western blots show only very small changes in protein levels, making it difficult to assess the functional consequences of such minor changes in expression or activity. Similarly, some of the in vivo effects are also rather modest.

Main points

The study focuses very heavily on PCK1 and I think it would be informative to also show the expression of other gluconeogenic enzymes, such as FBP-1, G6PC and PC. In some experiments, the changes in PCK1 expression are very modest, perhaps suggesting that additional factors are helping to change the response to p53 deletion. Could the authors trace the flux of labeled pyruvate to confirm the point at which gluconeogenesis fails in the p53 null mice/cells. Or does overexpression of PCK1 alone rescue the p53 null cells?

To support the importance of the subtle changes in p53 and PCK1 under various conditions, the authors should use direct metabolic tracing to provide evidence that these changes correlate with changes in gluconeogenesis.

I have a general concern with the p53 overexpression experiment. Acute induction of p53 generally leads to the activation of cell cycle arrest and cell death responses – possibly not reflecting the role of endogenous p53 levels in response to non-genotoxic stimuli such as those being examined here. Can the authors show that these canonical p53 responses are not being induced in the cells or tissue in which p53 ectopically expressed?

Specific points

Is the very modest accumulation or lack of accumulation of p53 with PUCNAc or OSMI-1 shown in Figure 1 enough the change p53 activity (eg p21 expression)?

In Figure 2, it would be useful to compare the activation of PCK1 to that of a well established p53 target gene, such as p21, and (in vitro) the effect of a strong p53 activating signal (such as Nutlin) on PCK1 expression.

The ChIP assay (Fig 2M) would benefit from a positive control of a known p53-regulated gene.

The effect of loss of liver p53 on blood glucose levels and rescue by re-expression of p53 is convincing (Figure 3), but the authors should show p53 expression levels in each of these conditions. Does overexpression of p53 lead to cell cycle arrest or cell death? Or altered liver function?

Figure 4B should show input p53 levels too.

Figure 4M and N – I don't think the authors can conclude that glucagon fails to stimulate blood glucose in hepatic p53 null mice – it still goes up in both models. Indeed, in Figure 4M the baseline glucose levels are lower in the p53 flox mice – if the increase in glucose is normalized to baseline is the difference still significant?

Figure 5 – do adrenaline or hydrocortisone increase liver PCK1 in vivo?

In both Figures 5 and 6 – are the changes in PCK1 protein levels accompanied by a change in activity?

What happens to p53 levels in the mice described in Figure 6? Is there any impact of p53 depletion on the insulin response? The model would suggest not.

The authors should state the number of mice used in each experiment in the Figure legends.

Reviewer #2:

Remarks to the Author:

In this paper, Gonzalez-Rellan et al. report that hepatic p53 is stabilized by O-GlcNAcylation upon starvation and regulates gluconeogenesis by regulating PCK1 transcription. They showed that glucagon, adrenaline and glucocorticoids increased protein levels of p53; while insulin decreased protein levels of p53. Perturbation of p53 levels impaired the ability of hepatocytes to regulate glucose levels. While it has been previously known that liver p53 levels are highly dynamic to changing nutrient environment, the authors were very detailed in the characterization of several mouse models with p53 perturbation. The novelty of the paper lies in two areas that could be further expounded: (1) O-GlcNAcylation of p53 is key to stabilizing the protein and aberrant O-GlcNAcylation of p53 is observed in type 2 diabetic patients; and (2) Different hormones and treatments directly perturb p53 protein levels via O-GlcNAcylation of p53.

Specific comments:

1. In Figure 1C, sugar was provided to fasted mice given that their circulating glucose level were severely affected. The authors could further clarify a few details. Namely, what type of sugar was provided? What were the amount and duration at which it was provided? Were there any controls (such as mannitol) that took into account of the osmotic pressure? A control for osmotic pressure should also be provided in Figure 1D when different concentrations of glucose were supplemented.
2. In Figure 1E-G, it was shown that O-GlcNAcylation of p53 was observed in starved cells and increased O-GlcNAcylation by PUGNAc treatment drove increased p53 protein expression, while decreased O-GlcNAcylation by OSMI-1 decreased p53 protein expression. These pharmacological interventions are highly non-specific, and can result in O-GlcNAcylation/de-O-GlcNAcylation of other pathways/proteins, thereby stabilizing/destabilizing p53. Yang et al. (Nat Cell Bio, 2007, 9(12); 1442) reported that O-GlcNAcylation of p53 at Ser149 has a stabilizing effect. I would suggest to carry out mutagenesis experiments on p53 to examine whether at starved conditions, p53 with S149A mutation would undergo increased or decreased expression. This would add credence that liver p53 protein levels are dependent of O-GlcNAcylation on its own.
3. Mutagenesis experiments can be performed to strengthen the argument that O-GlcNAcylation of p53 alters PCK1 levels as shown in Figure 2N/O. The p53 levels should be measured to demonstrate the efficacy of p53 silencing. In addition, the quantified results and representative western blots does not seem to match. In the representative western blot, the 1st and 3rd conditions look similar while the band in the 2nd condition looks much weaker.
4. In Figure 4B, the p53 IP experiment should include IgG control, and show the total amount of p53 that was pulled down.
5. Figure 4C, 5B and 5H, the authors should demonstrate that O-GlcNAcylation of p53 is important in glucagon/adrenaline/glucocorticoid action by carrying out p53 S149A mutagenesis in the THLE-2 line.
6. Figure 4M and 4N, genetic interaction between OGT and p53 in the animal models can be tested to ascertain that glucagon boosts blood glucose through the OGT-p53 axis.
7. In Figure 4, total CREB should be represented together with phosphor-CREB.
8. Figure 5A and 5G, do adrenaline and hydrocortisone treatments increase protein levels of p53 in vivo (mouse liver)?
9. Figure 6L, how does Akt regulate p53?
10. Figure 7, the levels of OGT should be measured in the livers of HFD mice with/without p53 depletion.
11. Individual data points should be better depicted for easier visualization. One notable example is Figure 2E.

Reviewer #3:

Remarks to the Author:

In this manuscript, the authors suggested a role of p53 in the regulation of hepatic gluconeogenesis. In particular, the authors showed that O-GlcNAcylation-dependent stabilization of p53 is essential in activating the transcription of PCK1, explaining the potential mechanism for p53-dependent increase in glucose production during fasting by mediating various hormonal signals such as glucagon, epinephrine and glucocorticoids. The hypothesis is interesting and may secure further considerations if the authors could provide the substantial amount of new experiments to support the current perspective.

Major comments

1. First and foremost, it is counter-intuitive to suggest that O-GlcNAcylation of a certain protein is enhanced upon starvation of glucose in the cultured cell. What is a potential precursor for UDP-N-acetylglucosamine during fasting (in vivo) or in KHH buffer (glucose-free media)? In fact, previous reports showed that O-GlcNAcylation of transcription factors or co-activators in the liver or hepatocytes (e.g. FOXO1, CRTC2, PGC-1 α , ChREBP) is enhanced in hyperglycemic conditions or with media containing high glucose levels (Housley MP, Rodgers JT, Udeshi ND, Kelly TJ, Shabanowitz J, Hunt DF, Puigserver P, Hart GW. *J Biol Chem.* 2008;283:16283-92; Housley MP, Udeshi ND, Rodgers JT, Shabanowitz J, Puigserver P, Hunt DF, Hart GW. *J Biol Chem.* 2009;284:5148-57; Dentin SHR, Xie J, Yates J, 3rd, Montminy M. *Science.* 2008;319:1402-1405; Guinez C, Filhoulaud G, Rayah-Benahmed F, Marmier S, Dubuquoy C, Dentin R, Moldes M, Burnol AF, Yang X, Lefebvre T, Girard J, Postic C.). The authors should detect whether these transcriptional regulators are indeed O-GlcNAcyated under starvation in vivo as well as in KHH buffer in cultured cells.
2. The authors suggested that p53 can only regulate transcription of PCK1 but not other gluconeogenic genes such as G6PC. The authors should determine whether overexpression/depletion of p53 affects hepatic glucose production with glycerol injection instead of pyruvate or lactate, since glycerol can bypass the requirement of PCK1 expression.
3. What is the specific mechanism by which glucagon, epinephrine, or glucocorticoid affects the level of O-GlcNAcylation of p53?
4. The authors should look for the longer-term effect of p53 knockout or p53 overexpression. The authors can utilize the current genetically modified mouse models for testing the chronic effect of hepatocyte-specific depletion/overexpression of p53 in vivo. These data are necessary to reinforce the current hypothesis.
5. In most western blot analysis, the authors showed the spliced gels instead of the intact gels (a rare exception is Fig 1E). The authors should provide the unspliced, original gels for each of the figures shown in the manuscript.
6. Some figures should be combined in one presentation. For example, Fig 2I should be combined to directly compare the effect of fasting and refeeding. Fig 6I should be also combined into one figure to directly compare the effect of p53 overexpression and the control in response to insulin.

Reviewer #1

Main points

The study focuses very heavily on PCK1 and I think it would be informative to also show the expression of other gluconeogenic enzymes, such as FBP-1, G6PC and PC. In some experiments, the changes in PCK1 expression are very modest, perhaps suggesting that additional factors are helping to change the response to p53 deletion. Could the authors trace the flux of labeled pyruvate to confirm the point at which gluconeogenesis fails in the p53 null mice/cells. Or does overexpression of PCK1 alone rescue the p53 null cells?

RESPONSE: We agree with the reviewer that this is a very interesting point. Other gluconeogenic enzymes (pPDH, PC, G6PC) were measured and as shown in Figure 2a, we do not detect changes in their expression, which was the reason to focus specifically on PCK1.

We agree with the Reviewer that in some experiments the changes in PCK1 expression are not huge, although we would not describe them as very modest since PCK1 levels reached significant differences and, more importantly, the biological consequences are in our view remarkable. Nonetheless, we totally agree that this part of the paper had to be strengthened. Thus, we performed the following additional experiments:

- a) PCK1 activity is stimulated in cells treated with adrenaline and this effect is blocked when p53 is silenced (**new Figure 6d**).
- b) PCK1 activity is stimulated in cells treated with hydrocortisone and this effect is blocked when p53 is silenced (**new Figure 6k**).
- c) PCK1 activity is inhibited in cells treated with insulin and this effect does not occur when p53 is overexpressed (**new Figure 7d**).
- d) To strengthen the link between p53 and PCK1, we have also measured the expression of both proteins in the liver of mice lacking leptin receptor (*db/db*), which are obese and diabetic. Interestingly, we found that protein levels of p53 and PCK1 are significantly higher in these mice compared to their controls (**new Figure 8j**). This result supports our findings in obese patients with type 2 diabetes, who also have increased levels of p53 and PCK1.
- e) We also performed mutagenesis experiments to demonstrate the key role of the O-GlcNAcylation of p53 to modulate PCK1 levels as requested by reviewer 2 (**new Figure 3**).
- f) As requested by the reviewer 3, we have also injected glycerol, which bypasses the requirement of PCK1 expression, in our two animal models lacking p53 in the liver (using a viral vector; and crossing p53 floxed mice with Alfp-Cre mice). In both cases, glycerol increased glucose levels at the same extent in both controls and hepatic p53 deficient mice (**new Supplementary figure 3d-4d**), indicating that contrary to pyruvate, glycerol does not require p53.

We thank the reviewer because these new results strongly support the importance of PCK1 for the effects of p53 on glucose metabolism.

We have performed an experiment tracing the flux of labeled pyruvate in Hep3B cells that do not express endogenous p53 using ¹³C(1)-pyruvate and monitored it by NMR spectroscopy. We transfected these cells with p53 WT and confirmed that the gluconeogenic route is overactivated, while the overexpression of mutated p53 (p53-

S149A) does not activate the route (**new Figure 3h**). Thus, these results reinforce our initial data suggesting that p53 induces gluconeogenesis, and more specifically, the *O*-GlcNAcylation of p53 in Serine 149 is essential for this gluconeogenic effect.

We have transfected THLE-2 cells silencing p53 (which lowers medium glucose levels) and another group of cells where in addition to silencing p53 we also overexpressed PCK1. As shown in the **new figure 2e** the overexpression of PCK1 does not just rescues but increases glucose levels. This is somehow expected since PCK1 is downstream p53, and thereby it seems logical that PCK1 can exert its gluconeogenic function independently of p53 inhibition.

To support the importance of the subtle changes in p53 and PCK1 under various conditions, the authors should use direct metabolic tracing to provide evidence that these changes correlate with changes in gluconeogenesis.

RESPONSE: Again, we appreciate the very interesting comment of the reviewer. In addition to the new experiments performed to corroborate the relevance of PCK1 as explained above, we have also performed new measurements of p21, a well-established p53 target gene to reinforce the changes in p53 levels (this is explained below in its correspondent specific point of reviewer 1). In this sense, we would like to highlight that it is accepted that moderate changes in the level of p53 maintain normal homeostasis of the liver, while a significant decrease, deletion or, on the other hand, chronic hyper-activation of p53 can lead to cancer initiation as summarized in the figure below taken from the review by Krstic J et al, Int J Mol Sci 2018:

Figure 2. The importance of p53 levels in liver physiology and pathology. Refer to the text for further details. Abbreviations: HCC, hepatocellular carcinoma (from PMID: 29558460).

I have a general concern with the p53 overexpression experiment. Acute induction of p53 generally leads to the activation of cell cycle arrest and cell death responses – possibly not reflecting the role of endogenous p53 levels in response to non-genotoxic stimuli such as those being examined here. Can the authors show that these canonical

p53 responses are not being induced in the cells or tissue in which p53 ectopically expressed?

RESPONSE: We agree with the reviewer that over-expressing a molecule can drive responses that do not necessarily reflect the function relating in the endogenous levels of that molecule. However, we think that to perform gain-and-loss-of-function experiments was an appropriate strategy to demonstrate the specificity of our results. In this case, both *in vitro* and *in vivo* results indicate that the overexpression of p53 in the liver increases PCK1 expression and glucose levels, which are the opposite effects to what we observed in cells and mice lacking p53 in the liver.

Nevertheless, we agree with the reviewer that aspects such as cell death and cell cycle deserved to be assessed. Therefore, we have now measured protein levels of markers of apoptosis (cleaved caspases 3 and 7) and immunostainings for Ki67 (cell cycle) in the livers of mice overexpressing p53. We found that protein levels of cleaved caspases 3 and 7 (**new Suppl. Figure 7**) as well as the immunostaining of ki67 remained unaltered in the two animal models where p53 was over-expressed (**new Suppl. Figure 7**). Overall, our data indicate that, at the studied times, our animal models do not display any change in cell cycle.

Specific points

Is the very modest accumulation or lack of accumulation of p53 with PUGNAc or OSMI-1 shown in Figure 1 enough the change p53 activity (eg p21 expression)?

RESPONSE: We have now measured the protein levels of p21 and detected that it was significantly upregulated by PUGNAc and downregulated by OSMI-1 (**new Supplementary figure 1e-f**).

In Figure 2, it would be useful to compare the activation of PCK1 to that of a well established p53 target gene, such as p21, and (in vitro) the effect of a strong p53 activating signal (such as Nutlin) on PCK1 expression.

RESPONSE: We agree with the reviewer on these points. As explained above, we found that KHH (medium without nutrients) stimulated p21 protein levels (**new Supplementary figure 1f**). In addition, we have also treated THLE-2 cells with doxorubicin, a well-established activator of p53, and found an increase in p53 protein levels as well as in PCK1 (**new Supplementary figure 2e**).

The ChIP assay (Fig 2M) would benefit from a positive control of a known p53-regulated gene.

RESPONSE: Again, the reviewer is right about this positive control. We have now measured p21 in the two experiments where ChIP assays were performed. *In vitro*, the overexpression of p53 caused an increase in p21 promoter expression (**new Supplementary figure 2g**). In the *in vivo* experiment, we also found higher p21 promoter expression levels after calorie restriction (**new Supplementary figure 2i**).

The effect of loss of liver p53 on blood glucose levels and rescue by re-expression of p53 is convincing (Figure 3), but the authors should show p53 expression levels in each of these conditions. Does overexpression of p53 lead to cell cycle arrest or cell death? Or altered liver function?

RESPONSE: we have now added the graphs shown p53 expression levels in the liver of p53 floxed mice and LKO mice with deletion or re-expression of p53 in their hepatocytes (**new Supplementary figure 5**).

In addition, we have assessed cell cycle arrest and cell death in our animal models. As shown in **new Supplementary figure 5**, the staining of cleaved caspase 3 and ki67 were undetectable compared with a positive control (hepatocellular carcinoma). Consistent with staining, hepatic protein levels of cleaved caspases 3 and 7 did not change among controls, p53 liver KO mice, or mice where p53 was rescued in the liver either in fed *ad libitum* or calorie restriction conditions (**new Supplementary figure 5**). Similar results were detected in mice where genetic knockdown of hepatic p53 was performed using virogenetic approaches, since staining of cleaved caspase 3 and ki67 did not show differences between groups (**new Supplementary figure 5**).

Figure 4B should show input p53 levels too.

RESPONSE: We have measured pulled-down p53 protein levels and did not find differences between KHH and KHH+glucagon (**new Figure 5n**).

Figure 4M and N – I don't think the authors can conclude that glucagon fails to stimulate blood glucose in hepatic p53 null mice – it still goes up in both models. Indeed, in Figure 4M the baseline glucose levels are lower in the p53-flox mice – if the increase in glucose is normalized to baseline is the difference still significant?

RESPONSE: The reviewer is totally right, and we have changed the text accordingly. The hyperglycemic effect of glucagon is partially blunted in mice lacking p53 in the liver. We have normalized glucose to baseline (please see rebuttal figure 1) and the differences are still significant.

Rebuttal figure 1. Normalized glucose levels in mice lacking p53 in the liver (left: using a viral vector; right: crossing p53 floxed mice with Alfp-Cre mice).

Figure 5 – do adrenaline or hydrocortisone increase liver PCK1 in vivo?

RESPONSE: We initially did not include these results because it was already published by other teams (see review PMID: 20826956) that both adrenaline and hydrocortisone stimulate PCK1 in liver. We have now performed these measurements and have reproduced the findings, with a significant increase in PCK1 protein levels after the injection of adrenaline and hydrocortisone. More importantly, we have combined mutagenesis experiments (required by reviewer 2) with the treatment of adrenaline and hydrocortisone to strength the relevance of *O*-GlcNAcylation of p53 on the actions of gluconeogenic hormones. More specifically, we now present the following results:

- In order to address the relevance of *O*-GlcNAcylation of p53 in glucagon and adrenaline action, we carried out p53 S149A mutagenesis in Hep3B cells treated with forskolin and found that, while it induced pCREB and p53, this compound

- was not able to increase PCK1 protein levels when p53 was mutated (**new Supplementary figure 8a**).
- b) Mice treated with adrenaline show increased protein levels of p53 and PCK1 in the liver (**new Supplementary figure 8e**).
 - c) Mice treated with glucocorticoids show increased protein levels of p53 and PCK1 in the liver (**new Supplementary figure 8g**).
 - d) Mice treated with adrenaline show augmented OGT levels in the liver (**new Supplementary figures 8f**).
 - e) Mice treated with glucocorticoids show decreased OGA levels (**new Supplementary figures 8h**).
 - f) Glucocorticoids are not able to increase PCK1 levels in cells transfected with p53 S149A (**new Supplementary figure 8h**). However, after the overexpression of p53-WT, glucocorticoids do increase PCK1 protein levels (**new Supplementary figure 8j**).

Taking together, this data evidence that *O*-GlcNAcylation of p53 is required for the actions of adrenaline and glucocorticoids.

In both Figures 5 and 6 – are the changes in PCK1 protein levels accompanied by a change in activity?

RESPONSE: We agree with the reviewer that this is an important question. We have now measured PCK1 activity in these experiments and the results are as follows:

- g) PCK1 activity is stimulated in cells treated with adrenaline and this effect is blocked when p53 is silenced (**new Figure 6d**).
- h) PCK1 activity is stimulated in cells treated with hydrocortisone and this effect is blocked when p53 is silenced (**new Figure 6k**).
- i) PCK1 activity is inhibited in cells treated with insulin and this effect does not occur when p53 is overexpressed (**new Figure 7d**).
- j) In addition, to strength the link between p53 and PCK1, we have also measured the expression of both proteins in the liver of mice lacking leptin receptor (*db/db*), which are obese and diabetic. Interestingly, we found that protein levels of p53 and PCK1 are significantly higher in these mice compared to their controls (**new Figure 8j**). This result supports our findings in obese patients with type 2 diabetes, who also have increased levels of p53 and PCK1.

What happens to p53 levels in the mice described in Figure 6? Is there any impact of p53 depletion on the insulin response? The model would suggest not.

RESPONSE: As shown in Figure 6h, protein levels of p53 are significantly elevated when we administered an adenoviruses overexpressing p53. Following suggestion, we have now analyzed hepatic p53 in mice treated with insulin. Our data shown that insulin diminishes p53 protein levels in the liver (**new Figure 7j**), which is consistent with decreased PCK1.

Regarding the second question, the administration of insulin in mice with p53 depletion does not have any impact at all. As shown in the former Supplementary figures 3c and 4c, ITT in the two animal models where p53 was depleted in the liver showed similar results to their controls.

The authors should state the number of mice used in each experiment in the Figure legends.

RESPONSE: We apologize for not including this information in the first version. This has been amended in the new version.

Reviewer #2

Specific comments:

1. In Figure 1C, sugar was provided to fasted mice given that their circulating glucose level were severely affected. The authors could further clarify a few details. Namely, what type of sugar was provided? What were the amount and duration at which it was provided? Were there any controls (such as mannitol) that took into account of the osmotic pressure? A control for osmotic pressure should also be provided in Figure 1D when different concentrations of glucose were supplemented.

RESPONSE: We appreciate all these relevant considerations.

- a) The sugar provided was sucrose. We have followed the protocol reported by Levine S & Saltzman A (Lab Anim 34(3):301-6, 2000).
- b) Sucrose was administered *ad libitum* under fasting conditions (24 hours). The reason to give sugar *ad libitum* was to provide enough amount to maintain liver weight and serum glucose levels, which was the idea of our experiment. Supplementation with lower amounts of sugar would result in a deficit that would likely affect liver homeostasis generating confounding factors.
- c) The question regarding the osmotic pressure is very interesting. In this sense, it is important to highlight that hyponatremia is not encountered when sucrose is given as cubes (Levine S & Saltzman A; Lab Anim 34(3):301-6, 2000), which is different to what has been found to the addition of glucose in drinking water. Furthermore, we have now performed new experiments in cells and mice to address this question. As shown in several figures of the manuscript, THLE-2 cells maintained in a medium without nutrients show higher protein levels of p53. In the new experiments, starved cells treated with mannitol still show higher p53 levels (**new Supplementary figure 1d**). In addition, mice fasted for 24h showed increased p53 protein levels, and fasted mice replaced with mannitol still had increased p53 (**new Supplementary figure 1c**).

These results indicate that mannitol, which has a potent osmotic diuretic action, does not modulate p53 levels, while the replacement with sugar decreases p53 protein levels in cells and mice. Overall, the data indicate that it is sugar but not changes in osmotic pressure what regulates p53 levels.

2. In Figure 1E-G, it was shown that O-GlcNAcylation of p53 was observed in starved cells and increased O-GlcNAcylation by PUGNAc treatment drove increased p53 protein expression, while decreased O-GlcNAcylation by OSMI-1 decreased p53 protein expression. These pharmacological interventions are highly non-specific, and can result in O-GlcNAcylation/de-O-GlcNAcylation of other pathways/proteins, thereby stabilizing/destabilizing p53. Yang et al. (Nat Cell Bio, 2007, 9(12); 1442) reported that O-GlcNAcylation of p53 at Ser149 has a stabilizing effect. I would suggest to carry out mutagenesis experiments on p53 to examine whether at starved conditions, p53 with S149A mutation would undergo increased or decreased expression. This would add credence that liver p53 protein levels are dependent of O-GlcNAcylation on its own.

RESPONSE: We thank the reviewer for this excellent comment. We got in contact with Prof. JW Cho, senior author of the mentioned study and with his help, we have designed a new set of additional experiments with a plasmid carrying out mutated p53 in Ser149 (p53-S149A).

- a) We first tested whether the overexpression of p53-S149A would result in an increase of protein levels of p53. We tested 3 different doses (2-4-8 $\mu\text{g}/\mu\text{l}$) in Hep3B cells, which do not express endogenous p53, and found that p53 levels are upregulated in a dose-dependent manner (**new Figure 3a**).
- b) Next, we transfected Hep3B cells with either p53 WT or p53-S149A and found that p53 WT increased PCK1 protein levels, while p53-S149A failed to modify PCK1 levels (**new Figure 3b**).
- c) We transfected Hep3B cells with either p53 WT or p53-S149A and found that p53 WT increased PCK1 activity, while p53-S149A failed to change PCK1 activity and it remained to baseline levels (**new Figure 3c**).
- d) We transfected starved Hep3B cells with p53-S149A and found that the mutant plasmid was able to increase p53 protein levels, however, under these conditions, the expression of PCK1 remained unaltered. Moreover, p53 with S149A mutation did not undergo increased accumulation during fasting (**new Figure 3d**).
- e) We transfected Hep3B cells with p53-S149A and treated them, with PUGNAc. We found that PUGNAc failed to modify p53 protein levels. Moreover, under these conditions, PUGNAc also fails to increase PCK1 protein levels. (**new Figure 3e**).
- f) We transfected THLE-2 cells with p53-S149A and found that the mutant plasmid did not increase PCK1 protein levels (**new Figure 3f**).
- g) We transfected THLE-2 cells with p53-S149A and contrary to the results found with p53 WT overexpression (Figure 2C), the mutant plasmid did not increase PCK1 activity (**new Figure 3g**).
- h) We have performed an experiment tracing the flux of labeled pyruvate in Hep3B cells that do not express endogenous p53 using $^{13}\text{C}(1)$ -pyruvate and monitored by NMR spectroscopy. We transfected these cells with p53 WT and confirm that the gluconeogenic route is overactivated, while the overexpression of mutated p53 (p53-S149A) does not activate the route (**new Figure 3h**).

Overall, these mutagenesis experiments together with our previous pharmacological data, clearly demonstrate that p53 effects on PCK1 levels are dependent of *O*-GlcNAcylation: while the transfection with the plasmid encoding p53-S149A increases p53 levels, this p53 is unable to activate PCK1.

3. Mutagenesis experiments can be performed to strengthen the argument that *O*-GlcNAcylation of p53 alters PCK1 levels as shown in Figure 2N/O. The p53 levels should be measured to demonstrate the efficacy of p53 silencing. In addition, the quantified results and representative western blots does not seem to match. In the representative western blot, the 1st and 3rd conditions look similar while the band in the 2nd condition looks much weaker.

RESPONSE: As explained above, new mutagenesis experiments have been carried out to demonstrate the relevance of *O*-GlcNAcylation of p53 to modulate PCK1 levels (**new Figure 3**). p53 levels have now been measured to demonstrate the efficacy of p53 silencing, and as expected, we found a significant decrease in THLE-2 cells transfected with si-p53 (**new Figure supplementary 2h**).

In addition to those, we have also transfected Hep3B cells with WT p53, and treated them with PUGNAc. Our results indicate that, after increasing p53 protein levels, PUGNAc is able to increase PCK1 levels (**new Figure 2p**, and the western blot has been repeated, in order to obtain a more representative western blot image, as the

reviewer suggested). Taken together, these results highlight the relevance of the O-GlcNAcylation on p53 gluconeogenic effects.

Moreover, p53 levels have now been measured to demonstrate the efficacy of p53 silencing, and as expected, we found a significant decrease in THLE2 cells transfected with si-p53 (**Supplementary figure 2h**).

4. In Figure 4B, the p53 IP experiment should include IgG control, and show the total amount of p53 that was pulled down.

RESPONSE: We agree with the reviewer and we now show total amount of pulled down p53 (**new Figure 5n**).

5. Figure 4C, 5B and 5H, the authors should demonstrate that O-GlcNAcylation of p53 is important in glucagon/adrenaline/glucocorticoid action by carrying out p53 S149A mutagenesis in the THLE-2 line.

RESPONSE: We agree with the reviewer that these are important questions. We have now performed additional experiments as follows:

- a) We transfected Hep3B cells with an empty plasmid or p53-S149A and treated them with vehicle or forskolin (PKA activator). We found that, as expected, in Hep3B control cells forskolin increases its surrogate marker pCREB but not p53 or PCK1 (since p53 is not present). When Hep3B cells were transfected with p53-S149A, forskolin still increased pCREB but it was not able to induce PCK1 despite p53 protein levels were augmented (**new Supplementary figure 8a**). Since PKA is downstream glucagon and adrenaline, this experiment applies to both signals (former Figures 4C and 5B).
- b) On the other hand, we transfected Hep3B cells with an empty plasmid or p53-S149A and treated them with vehicle or hydrocortisone. The results indicate that while p53-S149A increased p53 levels, it failed to increase PCK1 (**new Supplementary figure 8i**). In contrast, when we rescue p53 WT in Hep3B cells, the capacity of hydrocortisone to stimulate PCK1 protein levels is restored (**new supplementary Figure 8j**).

Taken together, these data indicate once again that, transfection with the plasmid p53-S149A increases p53 protein levels but this is not translated in an increase of p53 activation of gluconeogenesis, since the mutated form of p53 loses its capacity to regulate PCK1 and to mediate the stimulatory effect of glucagon, adrenaline and hydrocortisone on PCK1 levels.

6. Figure 4M and 4N, genetic interaction between OGT and p53 in the animal models can be tested to ascertain that glucagon boosts blood glucose through the OGT-p53 axis.

RESPONSE: We agree that this is an interesting question. To address it, we have pulled down p53 and then measured OGT protein levels. Our results indicate that OGT-p53 interaction is stimulated in the liver of mice treated with glucagon (**new Supplementary Figure 8d**).

7. In Figure 4, total CREB should be represented together with phosphor-CREB.

RESPONSE: We have now added total CREB to the figures (**new Figures 5c, 5e, 5f, 5i**).

8. Figure 5A and 5G, do adrenaline and hydrocortisone treatments increase protein levels of p53 in vivo (mouse liver)?

RESPONSE: We have now measured p53 protein levels in these samples and found that:

- a) p53 levels are upregulated in the liver of mice treated with glucagon but not in the liver of glucagon receptor KO mice treated with glucagon (**new Supplementary figure 8b-c**).
- b) p53 levels are increased in the liver of mice treated with adrenaline (**new Supplementary figure 8e**).
- c) p53 levels are increased in the liver of mice treated with hydrocortisone (**new Supplementary figure 8g**).

9. Figure 6L, how does Akt regulate p53?

RESPONSE: The fact that Akt regulates p53 has been described by others. In a well-described mechanism, Akt mediates the negative control of p53 levels through enhancing MDM2 (murine double minute 2)-mediated targeting of p53 for degradation. Accumulating evidence is beginning to suggest that, in certain circumstances, PTEN (phosphatase and tensin homologue deleted on chromosome 10)/PI3K/Akt also promotes p53 translation and protein stability, suggesting that additional mechanisms may be involved in the Akt-mediated regulation of p53 in tumors. For a review on this topic, please see PMID: 25109960.

10. Figure 7, the levels of OGT should be measured in the livers of HFD mice with/without p53 depletion.

RESPONSE: We have measured OGT in the following animal models:

- a) OGT protein levels are augmented in the liver of WT mice fed a HFD (60% fat) for 4 days (**new Figure 8a**).
- b) OGT protein levels remain unaltered in the liver of mice lacking p53 in the liver (animal model obtained after crossing Alfp-Cre with p53 floxed mice) fed a HFD (60% fat) for 4 days (**new Figure 8b**).
- c) OGT protein levels remain unaltered in the liver of mice lacking p53 in the liver (animal model obtained by virogenetic approaches) fed a HFD (60% fat) for 4 days (**new Figure 8c**).

11. Individual data points should be better depicted for easier visualization. One notable example is Figure 2E.

RESPONSE: We have widened the bars to facilitate the visualization of individual data points. We have also increased the size of the dots.

Reviewer #3

Major comments

1. First and foremost, it is counter-intuitive to suggest that O-GlcNAcylation of a certain protein is enhanced upon starvation of glucose in the cultured cell. What is a potential precursor for UDP-N-acetylglucosamine during fasting (in vivo) or in KHH buffer (glucose-free media)? In fact, previous reports showed that O-GlcNAcylation of transcription factors or co-activators in the liver or hepatocytes (e.g. FOXO1, CRTC2, PGC-1a, ChREBP) is enhanced in hyperglycemic conditions or with media containing high glucose levels (Housley MP, Rodgers JT, Udeshi ND, Kelly TJ, Shabanowitz J, Hunt DF, Puigserver P, Hart GW. *J Biol Chem.* 2008;283:16283–92; Housley MP, Udeshi ND, Rodgers JT, Shabanowitz J, Puigserver P, Hunt DF, Hart GW. *J Biol Chem.* 2009;284:5148–57; Dentin SHR, Xie J, Yates J, 3rd, Montminy M. *Science.* 2008;319:1402–1405; Guinez C, Filhoulaud G, Rayah-Benahmed F, Marmier S, Dubuquoy C, Dentin R, Moldes M, Burnol AF, Yang X, Lefebvre T, Girard J, Postic C.). The authors should detect whether these transcriptional regulators are indeed O-GlcNAcylated under starvation in vivo as well as in KHH buffer in cultured cells.

RESPONSE: We appreciate the important observation made by the reviewer. Indeed, it has been published that O-GlcNAcylation of transcription factors or co-activators in the liver or hepatocytes is enhanced in hyperglycemic conditions or with media containing high glucose levels. However, in recent years, it has become clear that metabolic regulation is far more complex and, in addition to being dependent on nutrient availability, **O-GlcNAcylation signaling is highly sensitive to various forms of cellular stress, including** for example, heat shock, hypoxia and **nutrient deprivation**. As a matter of fact, O-GlcNAcylation has been proposed to function as a nutrient and stress sensor that regulates cellular processes that range from transcription and translation to signal transduction and metabolism.

Glucose is taken up from the extracellular milieu by glucose transporters. Whereas the majority of glucose is used for glycolysis and glycogen synthesis, approximately 2-5% of glucose is channeled into the hexosamine biosynthetic pathway (HBP) where, through different steps, it is converted to UDP-GlcNAc, which can be attached to proteins by OGT.

This canonical pathway fits very well with finding an increase in O-glycosylated proteins in hyperglycemic conditions, as the reviewer suggests. However, another

phenomenon that has been consistently observed in other studies is an increase in cellular O-GlcNAcylation of specific proteins in conditions of nutrient deprivation (PMID: 18353774, PMID: 18174169, PMID: 19073609, PMID: 21526146). As the HBP flux is decreased under these conditions, other mechanisms should be involved. It has been suggested that one potential mechanism that could mediate this effect is the

called “salvage pathway”, by which some proteins lose its *O*-GlcNA moiety by the OGA activity, and free GlcNAc can be recycled and attached to new target proteins (please see scheme of the nutrient flux through the hexosamine biosynthetic pathway and the regulation of *O*-GlcNAcylation). Another potential mechanism that could explain the increase in *O*-GlcNAcylation during fasting is an upregulation of OGT expression, which may raise cellular *O*-GlcNAcylation levels despite the reduced intracellular availability of UDP-GlcNAc (PMID: 18353774, PMID: 18174169, PMID: 19073609, PMID: 21526146).

One example of the complexity of this post-translational modification is the *O*-GlcNAcylation of PGC1 α that shows a peak at 5 mM glucose and decreases gradually as glucose concentrations approached either hypo- or hyperglycemia (PMID: 22883232).

To note, the *O*-GlcNAcylation of specific proteins upon nutrient deprivation has been elegantly reviewed by Yang X & Qian K in Nat Rev Mol Cell Biol. 7:452-465; 2017.

All that being said, since nutrient availability regulates cellular *O*-GlcNAcylation levels non only by determining the abundance of UDP-GlcNAc but also by modulating the levels of *O*-GlcNAcylation enzymes, we measured OGT protein levels during fasting, and we found that OGT protein levels were significantly higher in the liver of mice fasted for 6, 12 and 24h (**new Figure 1e**).

2. The authors suggested that p53 can only regulate transcription of PCK1 but not other gluconeogenic genes such as G6PC. The authors should determine whether overexpression/depletion of p53 affects hepatic glucose production with glycerol injection instead of pyruvate or lactate, since glycerol can bypass the requirement of PCK1 expression.

RESPONSE: We appreciate the excellent comment of the reviewer. To reinforce the transcriptional regulation of PCK1 by p53, we have now measured PCK1 activity in several cellular and animal models as requested by reviewer 1 (former figures 5-6, which are now **new Figures 6 and 7**), as well as performed mutagenesis experiments to demonstrate the key role of the *O*-GlcNAcylation of p53 to modulate PCK1 levels as requested by reviewer 2 (**new Figure 3**). In addition, as requested by the reviewer, we have also injected glycerol in our two animal models lacking p53 in the liver (using a viral vector; and crossing p53 floxed mice with Alfp-Cre mice). In both cases, glycerol increased glucose levels at the same extent in both controls and hepatic p53 deficient mice (**new Supplementary figure 3d-4d**), indicating that contrary to pyruvate, glycerol does not require p53. We thank the reviewer because these new results strongly support the importance of PCK1 for the effects of p53 on glucose metabolism.

3. What is the specific mechanism by which glucagon, epinephrine, or glucocorticoid affects the level of *O*-GlcNAcylation of p53?

RESPONSE: Again, the reviewer raises an excellent point. As explained above, nutrient availability regulates cellular *O*-GlcNAcylation levels non only by determining the abundance of UDP-GlcNAc but also by modulating the levels of *O*-GlcNAcylation enzymes. And also as previously explained, OGT protein levels are significantly elevated in the liver of mice upon fasting. Thus, we have now measured OGT levels in the liver of mice treated with glucagon, adrenaline, or glucocorticoids, and the results are as follows:

- a) Glucagon increases hepatic OGT, but not OGA levels (**new Supplementary figure 8b**). This glucagon-induced effect is abolished in glucagon receptor KO mice (**new Supplementary figure 8c**).
- b) Adrenaline increases hepatic OGT, but does not affect OGA levels (**new Supplementary figure 8e**).
- c) Cortisol decreases hepatic OGA, but does not affect OGT levels (**new Supplementary figure 8h**).

These data are in keeping with the idea that glucagon and adrenaline exert their counterregulatory role in glucose homeostasis through a similar mechanism while the effect of glucocorticoids is different. Taken together our data further support the relevance of *O*-GlcNAcylation by p53 although further studies are needed to dissect out the molecular taxonomy involved.

4. The authors should look for the longer-term effect of p53 knockout or p53 overexpression. The authors can utilize the current genetically modified mouse models for testing the chronic effect of hepatocyte-specific depletion/overexpression of p53 in vivo. These data are necessary to reinforce the current hypothesis.

RESPONSE: We agree with the reviewer that longer-term measurements in p53 genetically modified mouse models would reinforce the current hypothesis. We have now performed a pyruvate tolerance test in 4, 5 and 6 month-old mice lacking p53 in the liver and we consistently found that the depletion of hepatic p53 ameliorates pyruvate-induced glucose levels (**new Supplementary figure 10a**). In addition, we also did PTTs in mice overexpressing p53 at the same time points, and in this case we detected that higher p53 levels in the liver are associated with increased glucose levels after pyruvate injection (**new Supplementary figure 10b**). Thus, these results are very similar to the ones obtained in 8–10-week-old animal models, and supports an important role of hepatic p53 in the synthesis of glucose across life-span/or at least up to adulthood.

5. In most western blot analysis, the authors showed the spliced gels instead of the intact gels (a rare exception is Fig 1E). The authors should provide the unspliced, original gels for each of the figures shown in the manuscript.

RESPONSE: Please note that unspliced original gels were provided as appendix figures. Images in original figures represent spliced gels because of space constraints but as usual, we are providing unspliced gels for all figures as appendix, labelling the samples represented in main the figures.

6. Some figures should be combined in one presentation. For example, Fig 2I should be combined to directly compare the effect of fasting and refeeding. Fig 6I should be also combined into one figure to directly compare the effect of p53 overexpression and the control in response to insulin.

RESPONSE: We understand the concern of the reviewer, but we have 5 different groups (fed, fast 6 h, fast 12 h, fast 24 h and refed) with a n=6 and thereby it is not possible to run all the samples within one gel. In addition, to run just 2-3 samples per group in 3 different gels is not very convenient for statistical analysis given the variability between gels. Those were the reasons to represent some data separately in Figure 2I. Just to get an idea of the effect of fasting at different time points, we have run one representative gel showing PCK1 protein levels, where it is possible to see that PCK1 levels increase in a time-dependent manner:

Rebuttal figure 2. Western blot showing PCK1 protein levels in the liver of mice upon fed ad libitum and fasted (6-12-24h).

For Figure 6I (**new Figure 7J-7K**), we have now run new gels showing protein levels of pAKT and PCK1 in the 4 groups (reducing the number of samples per group from 8 to 6). In these images, it is easily detected that insulin increases phosphorylated levels of AKT and decreases PCK1 in WT mice, while in mice overexpressing p53, insulin still phosphorylates AKT but loses its capacity to reduce PCK1.

Rebuttal figure 3. Western blot showing pAKT and PCK1 protein levels in the liver of mice (normal and overexpression of hepatic p53) treated with vehicle or insulin.

Reviewers' Comments:

Reviewer #1:

Remarks to the Author:

The authors have added further experiments to address the reviewers' comments and in general these do support the model. I would still prefer to see some tracing experiments to show directly the expected effects on gluconeogenesis.

A smaller point, I realise I don't understand the ChIP experiments (Figure 2m). The fragment 3 band seem to bind without any p53 (I assume this is p0) as Hep3B cells are null for p53. The binding here doesn't seem to be specific for p53.

Reviewer #2:

Remarks to the Author:

The authors have carefully considered this reviewer's suggestions and have introduced p53-S149A to the study. However, a couple of minor issues remain:

- a) The y-axis of Figure 3A should refer to p53 protein levels. In Figure 3A, the authors should clarify the labels as the amount of plasmids transfected but not dosage. It also seems that a further increase from 4 to 8 ug/ul of plasmids does not increase transfection efficiency. The authors could clarify the transfection efficiency here.
- b) The y-axis of Figure 3B should refer to PCK1 protein levels. This experiment, alongside Figure 3C, is useful in clarifying that O-GlcNAcylation of P53 at S149 is important in influencing PCK1 levels.
- c) The authors should include wildtype P53 in the experiments performed in Figure 3D and 3E to ascertain the effect of S149 O-GlcNAcylation on PCK1 expression.

The authors provide new data showing that transfection of Hep3B cells with the p53-S149A plasmid increased p53 protein levels but not p53 activity on gluconeogenesis, since the mutated form of p53 lost its capacity to mediate the stimulatory effect of glucagon, adrenaline and hydrocortisone on PCK1 levels. It would be important to include transfected wildtype p53 as a positive control, alongside p53-S149A and the negative control.

In new Supplementary Figure 8d, the western blots seem to be of rather low quality and can not be easily visualized.

The authors provide a point-by-point response to the question as to how Akt regulates p53. Such response should be put into the discussion to offer readers a clear context on which the experiments performed. The authors have described Akt as follows: "Altogether, these findings suggest that insulin activates AKT to diminish p53 that in turn down-regulates PCK1, which leads to lowering of blood glucose levels" (lines 428 to 430). This causal relationship suggested by the authors has not been fully validated, and need to be rephrased.

Reviewer #3:

Remarks to the Author:

The authors suggested that fasting-induced expression of OGT could be a major factor for the increased O-GlcNAcylation of specific proteins.

1. Could authors also found the similar changes in expression of OGT in cultured cells in KHH vs KHH+glucose?
2. Based on the previous reports and the current work, O-GlcNAcylation of transcriptional mediators such as FOXO1, PGC-1a, CRTC2, and p53 could impact upon hepatic gluconeogenesis. Could authors elaborate the relative importance or the contribution of each factor on this pathway, perhaps in the discussion?

Reviewer #1 (Remarks to the Author):

The authors have added further experiments to address the reviewers' comments and in general these do support the model. I would still prefer to see some tracing experiments to show directly the expected effects on gluconeogenesis.

REPLY: We agree with the reviewer that assessing the relevance of O-GlcNAcylated p53 to hepatic gluconeogenesis *in vivo* is of importance. Here, we would like to point out that there are 2 experiments showing the *in vivo* relevance of O-GlcNAcylated p53:

- a) **Figure 5n:** shows increased levels of O-GlcNAcylated p53 in the liver of mice treated with glucagon.
- b) **Figure 9e:** shows the positive correlation between hepatic expression of gfat1 and gfat2 (two key enzymes in the hexosamine pathway that produce UDP-GlcNAc) with the oral glucose tolerance test in patients.

That being said, we totally agree with the reviewer that we have not performed a large set of *in vivo* experiments compared to all *in vitro* results provided after the first revision.

According to the comment of the reviewer, the ideal experiment would be the administration of a viral vector encoding either p53 WT or the mutated p53 (p53-S149A) in p53 LKO mice, then inject $^{13-14}\text{C}(1)$ -pyruvate and finally to challenge those mice to nutrient deprivation. This would allow to trace the flux in conditions of stimulated gluconeogenesis. This is basically what we have done *in vitro* using Hep3B cells that do not express p53 (**Figure 3h**). However, although this is a technically feasible *in vitro* setting, the *in vivo* experiment is far of being straightforward.

From a technical point of view, tracing fluxes *in vivo* is indeed very complex. As a matter of fact, a very recent study has shown that fasted mice for 6, 12, and 18 hours use different gluconeogenesis substrates (PMID: 31918920). Although we would perform 24 hours fasting to follow the same experimental design than in Figure 1, such variability among only a few hours of fasting makes us to be afraid that mixing fasting plus virogenetic manipulations could lead to inconclusive data.

In addition to technical aspects, after an "unofficial" consultation to members of the Ethics Committee of USC, they have confirmed that a considerable long time might be needed to get an approval for a protocol where the same mouse must be: **i)** injected with a viral vector **ii)** plus injected with $^{13-14}\text{C}(1)$ -pyruvate **iii)** plus fasting.

For these reasons, we have done these alternative experiments:

Experiment 1. O-GlcNAcylation of p53 is significantly increased in the liver of mice upon fasting. This new data are now included as **new Figure 1f**.

Experiment 2. Hepatic p53 requires O-GlcNAcylation to up-regulate PCK1 or glucose levels.

In this experiment we knocked-down O-GlcNAc transferase (OGT), the enzyme that catalyzes the addition of the O-GlcNAc, in the liver of mice. In another group of mice, p53 was overexpressed after the inhibition of OGT. All the groups were finally placed on fasting for 24 hours (**new Figure 3i**). According to our *in vitro* data (**Figures 1g-1h-3e-5b-6i-7f**), in this *in vivo* experiment, we find that the inhibition of OGT down-regulates protein levels of p53 and PCK1 as well as blood glucose levels when compared to control fasted mice (**new Figures 3j-3k**). In mice with a subsequent overexpression of p53 in the liver, we detect that p53 protein levels are increased (**new Figure 3k**). However, this overexpression of p53 was not able to increase blood glucose levels neither PCK1 expression (**new Figure 3j**). Finally, immunoprecipitated O-GlcNAc is indeed decreased when OGT was knocked-down (**new Figure 3l**). In addition, the O-GlcNAc/p53 ratio is decreased after the inhibition of OGT, even when p53 is overexpressed (**new Figure 3l**).

In summary, these *in vivo* results show that p53 needs to be O-GlcNAcylated to control PCK1 and circulating glucose levels, because the overexpression of p53 under conditions where it is not able to suffer this post-transcriptional modification, it fails to regulate glucose metabolism. Overall, we are delighted to say that these new data totally support all our previous submitted data.

A smaller point, I realise I don't understand the ChIP experiments (Figure 2m). The fragment 3 band seem to bind without any p53 (I assume this is p0) as Hep3B cells are null for p53. The binding here doesn't seem to be specific for p53.

REPLY: The reviewer is totally right and we deeply apologize because there was a mistake in the identification of the cells used in this experiment. They were not Hep3B (human cells) but AML12 (mouse cells). The reason for this mistake was that ChIP assays were performed in both Hep3B and AML12 cells. The results representing Hep3B cells are shown below, and as the reviewer can see, there is no signal for p53, as one could expect, with the exception of fragment 3 where cells were transfected with pp53. These results were not included in the manuscript to avoid redundancy in a manuscript with already many panels.

Again, we are really sorry about this mistake that has been corrected throughout the manuscript and appreciate very much that the reviewer noticed it.

Rebuttal figure 1. ChIP assay in Hep3B cells.

Reviewer #2 (Remarks to the Author):

The authors have carefully considered this reviewer's suggestions and have introduced p53-S149A to the study. However, a couple of minor issues remain: a) The y-axis of Figure 3A should refer to p53 protein levels. In Figure 3A, the authors should clarify the labels as the amount of plasmids transfected but not dosage. It also seems that a further increase from 4 to 8 ug/ul of plasmids does not increase transfection efficiency. The authors could clarify the transfection efficiency here.

REPLY: We thank the reviewer for this comment. We have modified the y-axis accordingly. Regarding the transfection efficiency, it is true that the increase from 4 to 8 ug of plasmid does not augment the efficiency. According to our results, with 4 ug we reach the maximum transfection efficiency, that is why we do not find higher levels of p53 at higher amounts of plasmid.

b) The y-axis of Figure 3B should refer to PCK1 protein levels. This experiment, alongside Figure 3C, is useful in clarifying that O-GlcNAcylation of P53 at S149 is important in influencing PCK1 levels.

REPLY: We thank the reviewer for this correction, and we have modified the y-axis accordingly.

c) The authors should include wildtype P53 in the experiments performed in Figure 3D and 3E to ascertain the effect of S149 O-GlcNAcylation on PCK1 expression.

REPLY: Please note that we have included wildtype p53 in **Figures 2i and 2p**. In these figures, the overexpression of wildtype p53 significantly increase the protein levels of PCK1 (**Figure 2i**) and the rise of O-GlcNAc levels by PUGNac stimulates PCK1 levels in Hep3B cells when wildtype p53 is rescued but not with the empty plasmid (**Figure 2p**). Since the effects of wildtype p53 were shown in these figures, we did not include them in Figures 3D and 3E to avoid redundancy in a manuscript containing already a substantial amount of panels.

The authors provide new data showing that transfection of Hep3B cells with the p53-S149A plasmid increased p53 protein levels but not p53 activity on gluconeogenesis, since the mutated form of p53 lost its capacity to mediate the stimulatory effect of glucagon, adrenaline and hydrocortisone on PCK1 levels. It would be important to include transfected wildtype p53 as a positive control, alongside p53-S149A and the negative control.

REPLY: Please note that we have included transfected wildtype p53 as a positive control and its correspondent control in:

Figure 5j: Effect of forskolin on PCK1 protein levels in Hep3B cells before and after the rescue of wildtype p53.

Figure 5k: Effect of forskolin on glucose levels in Hep3B cells before and after the rescue of wildtype p53.

Similarly to glucagon, adrenaline also increases cAMP to exert its gluconeogenic activity, therefore we have not treated Hep3B cells with adrenaline. The treatment with forskoline, which raises cAMP levels, was meant to explain the mechanism of both glucagon and adrenaline.

Supplementary figure 8j: Effect of hydrocortisone on PCK1 protein levels in Hep3B cells before and after the rescue of wildtype p53.

In new Supplementary Figure 8d, the western blots seem to be of rather low quality and can not be easily visualized.

REPLY: We apologize if the quality of the photos was low. We have not put them at higher resolution and hope they can be easily visualized.

The authors provide a point-by-point response to the question as to how Akt regulates p53. Such response should be put into the discussion to offer readers a clear context on which the experiments performed. The authors have described Akt as follows: “Altogether, these findings suggest that insulin activates AKT to diminish p53 that in turn down-regulates PCK1, which leads to lowering of blood glucose levels” (lines 428 to 430). This causal relationship suggested by the authors has not been fully validated, and need to be rephrased.

REPLY: We totally agree with the reviewer and have now included a brief discussion on this point. We have also rephrased the sentence.

Reviewer #3 (Remarks to the Author):

The authors suggested that fasting-induced expression of OGT could be a major factor for the increased O-GlcNAcylation of specific proteins.

1. Could authors also found the similar changes in expression of OGT in cultured cells in KHH vs KHH+glucose?

REPLY: We have now performed this western blot, and as shown in **new Figure 1d**, protein levels of OGT are significantly increased in starved cells (KHH) and return to baseline levels after glucose supplementation.

2. Based on the previous reports and the current work, O-GlcNAcylation of transcriptional mediators such as FOXO1, PGC-1a, CRTC2, and p53 could impact upon hepatic gluconeogenesis. Could authors elaborate the relative importance or the contribution of each factor on this pathway, perhaps in the discussion?

REPLY: We have now briefly discussed the potential contribution of the O-GlcNAcylation of these transcriptional mediators.

Reviewers' Comments:

Reviewer #1:

Remarks to the Author:

The authors have addressed my concerns and I think the paper is now ready for publication.

Reviewer #2:

Remarks to the Author:

The concerns raised by this reviewer have been satisfactorily addressed. I have no further comments.

Reviewer #3:

Remarks to the Author:

The authors adequately addressed the points that were raised by the reviewer in the previous version of the manuscript.